# Mastering Sparse CUDA Generation through Pretrained Models and Deep Reinforcement Learning

**Yaoyu Wang**[12][*], **Hankun Dai**[2][*], **Zhidong Yang**[123], **Junmin Xiao**[12], **Guangming Tan**[12][†]
[1]State Key Lab of Processors, Institute of Computing Technology, Chinese Academy of Sciences
[2]University of Chinese Academy of Sciences [3]Hong Kong University of Science and Technology
`wangyaoyu21s@ict.ac.cn`, `daihankun19@mails.ucas.ac.cn`,
`kezdyang@ust.hk`, `{xiaojunmin,tgm}@ict.ac.cn`

## Abstract

Code generation is a crucial research area in the field of artificial intelligence, holding the potential to revolutionize software development and streamline programming processes. However, generating the high-performance code, which need to be executed in a shorter time for the low-latency scenario, remains a formidable challenge. Existing methods often struggle to account for the irregularity of input sparse data in sparse programs and the need for domain-specific architectural knowledge, leading to sub-optimal performance. To tackle these issues, we propose the SparseRL framework. SparseRL leverages deep reinforcement learning, treating a pre-trained language model as a stochastic policy. It takes the row and column indices of non-zero elements in the sparse matrix as input and generates CUDA code as output for sparse matrix operations. We also introduce a domain-specific code generation mechanism for the dynamic input, a sinusoidal embedding technique tailored for sparse matrices, and a hierarchical reward function that considers both code correctness and execution efficiency. Experimental results demonstrate SparseRL achieves state-of-the-art performance. In sparse matrix-vector multiplication (SpMV) tasks, it improves the compilation rate by 20% compared to existing methods, and the generated code runs 30% faster on average. For sparse matrix-dense matrix multiplication (SpMM) tasks, SparseRL also shows significant performance gains. These results highlight the effectiveness of SparseRL in generating high-performance CUDA code for sparse matrix operations.

## 1 Introduction

In recent years, code generation has witnessed a remarkable advance with the increasing adoption of deep learning and neural language models. Extensive research aims to generate a sequence of code as the output program according to various code-related targets such as code completion (Li et al., 2017; Wei et al., 2023), code translation (Zhu et al., 2022; Pan et al., 2023), and program synthesis (Li et al., 2022; Le et al., 2022). While these methods achieve promising results, we observe that they still fail to generate high-performance code, which needs to be executed in a shorter time to meet low-latency scenario requirements. For example, in large language model (LLM) inference and graph neural network (GNN) computation, users expect to obtain results more quickly. However, manual optimization is very time-consuming (e.g., it takes several years from the emergence of transformer (Vaswani et al., 2017) to the appearance of Flash-attention (Xia et al., 2023)).

The difficulty of generating high-performance computing (HPC) code stems from two major factors. First, in the context of code generation by LLMs, the tasks often involve sparse programs, which differ from dense computations with fixed input patterns. For these sparse programs, execution patterns for data access and computation are dynamic and closely tied to the input sparse data (e.g., irregular and dynamic graph structures in GNNs) and can only be determined at runtime (Kim et al., 2024; Sadman & Qasem, 2025), requiring customized implementations to achieve minimal execution time

---

[*]Equal contribution. Our code is available on `https://github.com/QiWu-NCIC/SparseRL`.
[†]Corresponding author.

for varying input sparse structures (Du et al., 2022; Li et al., 2013). Second, effective performance optimization demands domain-specific architectural expertise. Many prior works (Liu & Vinter, 2015; Naumov et al., 2010) have relied on manual knowledge with performance optimization, such as improving cache hit rates (Gómez et al., 2021; Yan et al., 2014) and load balance (Liu & Vinter, 2015; Merrill & Garland, 2016), to enhance program execution efficiency.

Among the various tasks requiring high-performance code generation, sparse matrix-vector multiplication (SpMV) is a fundamental and critical computational operation. It has been widely applied in pruned LLM inference (Li et al., 2024; Xu et al., 2024) and GNN computation (Yao et al., 2023; Jia et al., 2022). Due to the performance impact of the input sparse matrix structure in sparse programs, the goal is to generate high-performance SpMV CUDA code for each dynamic input sparse matrix. There has been numerous research (Li et al., 2013; Yan et al., 2014; Zhao et al., 2018) of SpMV on GPUs and there is no one-size-fits-all solution for implementing high-performance SpMV code on GPUs (Zhao et al., 2018; Su & Keutzer, 2012), because different types of sparse matrices require tailored implementations to achieve the optimal performance.

Additionally, the generated CUDA code for SpMV should guarantee two conditions. First, the generated code should be syntactically and functionally correct to pass compilation and deliver correct computational results. Second, it must also be optimized for high-performance execution on the GPU to achieve shorter execution time. However, current methods face three major limitations.

_**First**_, current methods use the conventional supervised next-token prediction objective (Bengio et al., 2015; Ranzato et al., 2015) to train the model, which maximizes the likelihood of the next ground-truth token. For example, in the pre-training and fine-tuning of models, token-level matching objectives like cross-entropy loss (Papineni et al., 2002; Ren et al., 2020) are commonly employed. For generating high-performance SpMV CUDA code, this supervised approach is insufficient, because multiple semantically correct SpMV programs (ground-truths) exist, but only a few can achieve the optimal performance for each sparse matrix.

_**Second**_, current methods ignore the crucial rewards about the execution efficiency of generated code. This directly impacts the code's quality measured by the execution time of code. Existing methods neglect this crucial reward during the model optimization and generation processes. Incorporating execution efficiency (time) into the learning objectives during model optimization could better align with the ultimate goal of generating semantically correct and high-performance CUDA code.

_**Third**_, due to the impact between performance and input sparse data, a customized program is required for each sparse data. However, existing LLMs face inherent challenges in narrowing the modality gap between input sparse data representations and CUDA code generation. This mismatch necessitates the adaptation to translate input sparse data into meaningful information for the model to generate a customized program.

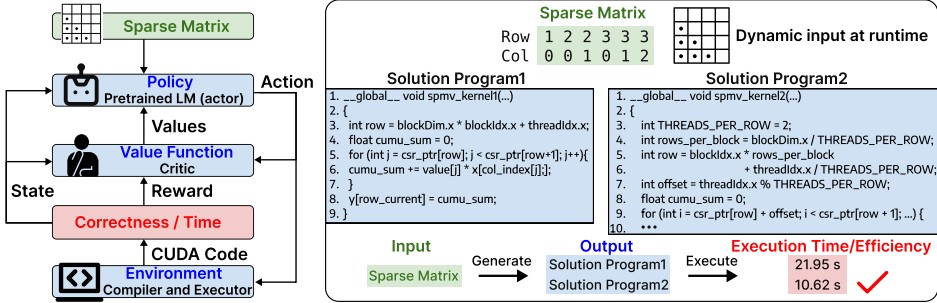

Figure 1: **An example task (Right)**: Each task is defined by a sparse matrix, which contains row and column indices of non-zero elements. The expected output is an SpMV program to be checked for functional correctness and execution efficiency. The input sparse matrix is the dynamic input at runtime and closely related to execution time. **A high-level overview of our SparseRL method for sparse CUDA code generation (Left)**: we treat a pre-trained language model (LM) as a stochastic policy, code generation steps as actions, and rewards can be estimated based on the functional correctness and execution time of output programs from the compiler and executor (environment).

To address the above issues, we propose a novel deep reinforcement learning-based SpMV CUDA code generation method called SparseRL (Figure 1), aiming at generating high-performance and

correct CUDA code. The SparseRL is built upon the idea of treating a pre-trained LLM for code generation as a stochastic policy. The input of the LLM in SparseRL is the row and column indices of non-zero elements in the sparse matrix and the output is the generated SpMV CUDA code. The process of CUDA code generation is regarded as a sequence of actions. Each action corresponds to the generation of a token in the CUDA code. The utilization of DRL allows the model to learn from the non-differential metrics of the compiler and executor (environment). The environment checks the syntactic and functional correctness of the generated CUDA code and provides information about the execution time of the program, which is used to calculate rewards. The main contributions of SparseRL can be summarized as follows:

- We propose a domain-specific code generation framework for the dynamic input of sparse matrix operators, which is based on DRL and enables the feedback from the compiler and executor to optimize the code generation process.
- We propose a row/column indices-based sinusoidal embedding technique for non-zero elements in sparse matrices, which directly captures the dynamic input sparse matrix structure to adapt the execution patterns at runtime.
- We devise a novel hierarchical reward, which incorporates execution time and compilation success as runtime and compile-time information into the model to generate better code.

The experiments prove that our SparseRL achieves state-of-the-art (SOTA) results on generating high-performance CUDA code compared with other methods. In fact, our method is not limited to the task of SpMV. We use it as an example to show how to incorporate rich multimodal information (e.g. runtime information of the program) to design a more powerful expert. So as a verification, we extend our method to the sparse matrix-dense matrix multiplication (SpMM) task and also achieve significant improvement, which validates the generalization and transferability of our SparseRL.

## 2 RELATED WORK

### 2.1 PRE-TRAINED MODELS FOR CODE GENERATION

Pre-trained models for code generation have evolved with transformer-based models like Code-BERT (Feng et al., 2020). Then, large-scale autoregressive models such as AlphaCode (Li et al., 2022) have emerged to enhance the model's ability of natural-language-to-code translation and code completion. Retrieval-augmented models like CodeT5 (Wang et al., 2021) and StarCoder (Li et al., 2023) further refine task-specific objectives. Despite above advancements, challenges remain in handling code correctness and low-latency demand. These challenges inspire the research on reinforcement learning (Le et al., 2022; Shojaee et al., 2023) for improving the code quality. However, SparseRL encodes the sparse matrix to generate high-performance CUDA code. This type of input modality is similar to AlphaFold (Jumper et al., 2021), which also leverages a modality distinct from vision and text. Our work is also different from code implementation of algorithm problems and dense computation (elaborated in Appendix A.1). We also have a more direct engagement with contemporary research on LLM-based generation of high-performance code in Appendix A.6.4.

### 2.2 ARTIFICIAL PROGRAMMING FOR SPMV ON GPUS

Sparse matrix and vector multiplication (SpMV) operation multiplies a sparse matrix by a dense vector, and obtains a dense vector as the result. It is a fundamental operation that plays a crucial role in a wide range of numerical simulation applications (Shantharam et al., 2011; Abdelfattah et al., 2015), and artificial intelligence domains (Wang et al., 2019; Mao et al., 2017). The challenge of designing SpMV on GPUs mainly arises from the highly diverse distribution of non-zero elements in sparse matrices (Li et al., 2013; Zhao et al., 2018). Different CUDA code needs to be implemented for different sparse matrices to achieve high-performance execution (shorter execution time), where examples are in Appendix A.10.

The history of optimizing SpMV has lasted for a long time (for example, CSR (Bell & Garland, 2008) → CSR5 (Liu & Vinter, 2015) → DASP (Lu & Liu, 2023) and has always been a hot and popular topic (e.g., CSR-Adaptive (Greathouse & Daga, 2014),ACSR (Ashari et al., 2014), Merge-based (Merrill & Garland, 2016) and etc. ). Over the past few decades, substantial efforts have been dedicated to enhancing the performance of SpMV on GPUs. These ways of artificial programming for SpMV are based on observed matrix properties and are specifically devised for the characteristics of the underlying hardware. However, these methods are often manually crafted and lack the adaptability to handle diverse matrix types and changing hardware environments. Additionally, they do not leverage

the emerging LLM techniques, which could potentially offer more flexible and intelligent solutions for SpMV code implementations. We strengthen positioning against prior work in Appendix A.26.

### 2.3 REINFORCEMENT LEARNING FOR SEQUENCE GENERATION

Reinforcement learning (RL) has been applied to optimize non-differentiable metrics in sequence generation tasks. For instance, the REINFORCE algorithm (Williams, 1992) has been used to improve BLEU (Papineni et al., 2002) and ROUGE (Lin, 2004) scores in translation and summarization models. More recently, InstructGPT (Ouyang et al., 2022) introduced a Reinforcement Learning from Human Feedback (RLHF) fine-tuning procedure. When it comes to code generation, RL has also been employed to enhance the quality of generated code (Le et al., 2022; Dou et al., 2024).

Despite these efforts, existing RL-based methods for Sparse CUDA code generation have limitations. They struggle to generate high-performance code for low latency scenario, and leverage the complex relationship between input sparse data and execution patterns at runtime.

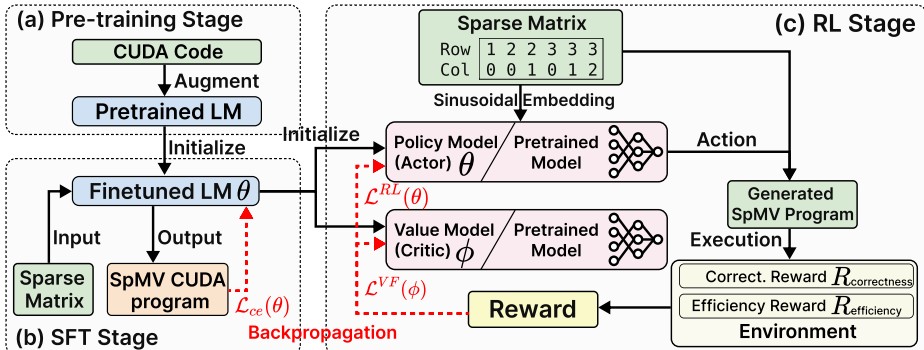

Figure 2: **Overview of our method to optimize pretrained LMs for sparse CUDA code generation**. (a) At pre-training stage, additional CUDA code is used to augment the LM. (b) At supervised fine-tuning (SFT) stage, the LM is finetuned for the Sparse CUDA code generation. (c) At RL stage, the actor and critic networks are first initialized from the finetuned LM, and then updated based on the reward of RL. The reward function is composed of correctness and efficiency rewards.

## 3 SPARSERL

### 3.1 TASK DESCRIPTION

We take the SpMV task as a sparse CUDA code generation example. Following a sequence-to-sequence approach, the SpMV CUDA code generation task contains a sparse matrix as an input sequence $X = ((r_1, c_1), \cdots, (r_N, c_N))$ and an output sequence of SpMV CUDA code $\hat{Y} = (\hat{y}_1, \cdots, \hat{y}_T), \hat{y}_t \in \mathcal{V}$, where $r_i$ is the row index and $c_i$ is the column index of the $i$-th non-zero element, and $N$ is the total number of non-zero elements. The output of each decoding step $t$ is a distribution over the vocabulary $\mathcal{V}$, computed by the Softmax function $\hat{y}_t = \text{Softmax}(\text{Linear}(h_t))$ where $h_t$ is the contextual hidden state at decoding step $t$.

Conventionally during fine-tuning process, model parameters ($\theta$) are learned by maximizing the likelihood of the ground-truth target code. Let the set of all possible SpMV ground-truth code be $\mathcal{Y} = \{Y_1, Y_2, \cdots, Y_n\}$, where $n$ represents the number of ground-truth code, and $n > 1$. This indicates that there exists multiple CUDA code that can correctly perform SpMV operation. Denoting $\hat{Y} = (\hat{y}_1, ..., \hat{y}_T)$ as the ground-truth code, the objective of supervised fine-tuning (SFT) stage is to minimize the cross-entropy loss: $\mathcal{L}_{ce}(\theta) = -\sum_t \log p_\theta(\hat{Y}|X) = -\sum_t \log p_\theta(\hat{y}_t|\hat{y}_{1:t-1}, X)$, where the conditional probability $p_\theta$ is parameterized following the above softmax function. The model generates sequences of CUDA SpMV code by autoregressively sampling token $\hat{y}_t$ from the distribution $p_\theta(\cdot|\hat{y}_{1:t-1}, X)$. In the context of SpMV, the model is evaluated based on both correctness and efficiency of generated CUDA code. Each test includes an input and multiple ground-truth outputs of different execution efficiencies. Let the execution time of the code $\hat{Y}$ for sparse matrix $X$ be $E(\hat{Y}|X)$. Our goal is to generate the code $\hat{Y}$ that meets the following two conditions, which can be represented by the formulas:

$$\begin{cases} \text{Compile}(\hat{Y}) = \text{True} \\ \text{Correct}(\hat{Y}, X) = \text{True} \\ E(\hat{Y}|X) \le E(Y_i|X), \forall Y_i \in \mathcal{Y} \end{cases} \quad (1)$$

where $\mathcal{Y}$ denotes all functionally correct SpMV CUDA programs (not limited to pre-defined ground-truths). It should be noted that the execution time $E(Y_i|X)$ is closely related to the sparse matrix $X$, because different sparse matrix input $X$ require different CUDA code $\hat{Y}$ to achieve optimal execution efficiency.

To address this task, our work contains the following essential modules as shown in Figure 2. ***First***, at pre-training stage, we augment the pre-training data with a large amount of CUDA code. By incorporating this domain-specific training data, the model can learn the specific programming paradigms and optimization techniques for CUDA-based sparse code. ***Second***, during SFT and RL stages, we replace the input of the language model with the row and column indices of each non-zero element of the sparse matrix, and output the high-performance CUDA code for SpMV, so the model can learn how to adapt the code generation process based on the distribution of non-zero elements from the input sparse matrix. ***Third***, at RL stage, we further use RL to optimize the model and design a novel reward for execution efficiency, which aims at fine-tuning the pre-trained model to better suit the SpMV CUDA code generation task.

## 3.2 PRETRAINING AND FINE-TUNING LANGUAGE MODELS

**Pre-training stage with CUDA code augmentation.** We augment the pre-training data with a large amount of CUDA code, because the CUDA code involves unique requirements such as parallel computing, memory management on GPUs, and optimized algorithms for specific hardware architectures. We follow the pre-training setup in CodeT5 (Wang et al., 2023a; 2021), which has demonstrated remarkable capabilities in handling programming languages. The augmented (CUDA code) dataset includes hand-crafted high-performance SpMV code (Liu & Vinter, 2015; Zhao et al., 2018; Du et al., 2022; Merrill & Garland, 2016) and other CUDA programming languages on GitHub by reserving only permissively licensed code ("mit", "apache-2", "bsd-3-clause", "bsd-2-clause", "cc0-1.0", "unlicense", "isc").

**SFT and RL stage with sparse matrix embedding.** Different from previous LLMs typically relying on word embeddings and positional encodings, we devise a sparse matrix embedding strategy to input the sparse matrix to the model. We utilize the row and column indices of the non-zero elements in the sparse matrices during the fine-tuning process. Let the sparse matrix be with dimensions $m \times n$, and denote the set of non-zero elements' row-column indices as $X = \{(r_i, c_i)\}_{i=1}^{N}$, where $r_i$ is the row index and $c_i$ is the column index of the $i$-th non-zero element, and $N$ is the total number of non-zero elements. We leverage the sinusoidal embedding (Vaswani et al., 2017) to normalize the input row and column indices (theoretical foundation is in Appendix A.2 and ablation study is in Section 5.2), which is formulated as:

$$PE_{(ind,2j)} = \sin\left(ind/(10000^{2j/d_{model}})\right), PE_{(ind,2j+1)} = \cos\left(ind/(10000^{2j/d_{model}})\right) \quad (2)$$

where $ind$ represents the index value (either the row or column index in our SparseRL), $j$ is the dimension index within the embedding vector, and $d_{model}$ is the embedding vector's dimension.

For the row indices, we calculate the sinusoidal embedding vectors. Given a row index $r_i$, the sinusoidal-encoded row vector $\mathbf{e}_{r_i}$ of dimension $d_{model}$ is obtained by applying the above formula 2 with $ind = r_i$. Similarly, for the column index $c_i$, the sinusoidal-encoded column vector $\mathbf{e}_{c_i}$ of dimension $d_{model}$ is calculated using $ind = c_i$. We then concatenate the sinusoidal-encoded row and column vectors for each non-zero element. That is, for the $i$-th non-zero element, we form a vector $\mathbf{e}_i = [\mathbf{e}_{r_i}|\mathbf{e}_{c_i}]$. These vectors $X' = \{\mathbf{e}_i\}_{i=1}^{N}$ are used as the input to the model. During the SFT and RL stages, the language input prompt is removed and only sparse matrices are provided as input. The target output is the SpMV CUDA program. For converting the modality of input sparse matrix to language modality, an additional linear layer is applied to map the concatenated vectors to the appropriate dimension. This strategy enables the model to directly capture the structural information of the sparse matrix from the non-zero element indices.

### 3.3 RL Problem Formulation

The code generation procedure can be formulated as a sequential discrete finite-horizon Markov decision process (MDP) with the use of RL in which an agent interacts with the environment over discrete steps $T$, and we employ the PPO algorithm (Schulman et al., 2017) training details are in Appendix A.5). In SparseRL, the RL problem is formulated as follows:

**State** $\mathcal{S}$: The state at time $t$ is $s_t = (\hat{y}_{1:t-1}, X)$ ($s_t \in \mathcal{S}$), determined by the sparse matrix $X$ (containing row/column indices of non-zero elements) and the partial code sequence $\hat{y}_{1:t-1}$ generated before $t$, providing context for next token generation.

**Action** $\mathcal{A}$: The action at time $t$ is $a_t = \hat{y}_t$ ($a_t \in \mathcal{A}$), corresponding to the generated CUDA code token. Each token in the vocabulary ($y_t \in \mathcal{V}$) represents a potential action.

**Policy** $\pi_\theta(a_t|s_t)$: A stochastic policy network (parameterized by $\theta$, initialized from the SFT-finetuned model) predicts the next token based on $\hat{y}_{1:t-1}$ and $X$. The action $a_t$ is selected via top-$k$ sampling, yielding the full SpMV CUDA code $\hat{Y} = (\hat{y}_1, \cdots, \hat{y}_T)$ at episode end $T$.

### 3.4 Reward Function

The final reward $R_{\text{final}}(\hat{Y}, X)$ is calculated at the end of the code generation episode, integrating the generated code's syntactic correctness, execution efficiency, and memory usage.

#### 3.4.1 Correctness Reward

The correctness reward ($R_{\text{correctness}}$) follows a hierarchical structure, combining compilation and test results:

**Compilation reward** ($R_{\text{compile}}$): A reward of $+0.5$ is given if the generated CUDA code compiles successfully; otherwise, a reward of $-0.5$ is assigned.

**Test reward** ($R_{\text{test}}$): This reward is only considered if compilation succeeds. A reward of $+0.5$ is granted if the compiled code passes SpMV functional tests; otherwise, $-0.5$ is assigned.

**Total correctness reward**: The sum of the compilation reward and (if compilation succeeds) the test reward, formally expressed as $R_{\text{correctness}} = R_{\text{compile}} + \mathbb{I}_{\text{compile}} \cdot R_{\text{test}}$, where $\mathbb{I}_{\text{compile}}$ is an indicator function that equals 1 for successful compilation and 0 otherwise.

#### 3.4.2 Execution Efficiency Reward

The efficiency reward ($R_{\text{efficiency}}$) is activated only when the code is both compilable and functionally correct. It is calculated as a scaled value of the performance improvement over a baseline (cuSPARSE library), defined as $R_{\text{efficiency}} = r_{\text{eff}} \times \left( \frac{t_{\text{base}}(X)}{t(\hat{Y}, X)} - 1 \right) \cdot \mathbb{I}_{\text{test}}$, where $t_{\text{base}}(X)$ is the baseline execution time (measured by cuSPARSE (Naumov et al., 2010) library provided by NVIDIA), and $t(\hat{Y}, X)$ is the execution time of the generated code $\hat{Y}$ for the given sparse matrix $X$, and $\mathbb{I}_{\text{test}}$ is an indicator function that is 1 if all tests pass and 0 otherwise. $r_{\text{eff}}$ is a scaling factor that determines the importance of the execution efficiency in the overall reward. We use the average execution time of 1000 iterations to prevent fluctuations. The normalization details of $R_{\text{eff}}$ are in Appendix A.4.2.

#### 3.4.3 Overall Reward

The final reward integrates correctness, efficiency, and a memory penalty: $R_{\text{final}}(\hat{Y}, X) = R_{\text{correctness}} + R_{\text{efficiency}} - r_{\text{penalty}} \cdot \mathbb{I}_{\text{memory}}$. Here, $\mathbb{I}_{\text{memory}}$ is an indicator function (1 if the code exceeds a predefined memory limit, 0 otherwise), and $r_{\text{penalty}}$ is the penalty for excessive memory usage (Appendix A.25). Sensitivity study of $r_{\text{eff}}$ and $r_{\text{penalty}}$ in reward hyperparameters is in Appendix A.4.1, and the balance of $R_{\text{efficiency}}$ and $R_{\text{correctness}}$ is in Appendix A.4.3.

### 3.5 Objective

Our objective is to find an optimal set of model parameters such that the generated program for a given matrix can meet the requirements of correctness and efficiency. During the fine-tuning process, we first use supervised fine-tuning (SFT), and then use RL to optimize the network model. In addition, for early stopping the generation of error code during the generation process, we integrate a dynamic syntax correctness verification mechanism during the decoding process and use the code extraction tool similar to the public QwenLM repository (Appendix A.3).

## 4 EXPERIMENT

### 4.1 EXPERIMENTAL SETTINGS

We use two machines, and each machine is equipped with 8 Tesla V100/A100 GPUs over a duration of five days. The Adam optimizer (Kingma & Ba, 2014) is employed, configured with a learning rate of $1 \times 10^{-4}$. The experiment of balance between exploration and exploitation is in Appendix A.9.

**Benchmarks.** We compare SparseRL with two types of works. *(1) Code Generation:* (i) Qwen3 (Hui et al., 2024; Yang et al., 2025), (ii) DeepSeek-R1 (Guo et al., 2025), (iii) CodeRL (Le et al., 2022), (iv) PPOCoder (Shojaee et al., 2023). *(2) Artificial Library:* (i) cuSPARSE (v12.1) library (Naumov et al., 2010), which is provided by NVIDIA. (ii) TVM-S (Chen et al., 2018) is the sparse version of TVM. *(3) Closed-source models:* GPT-o3-pro, GPT-5, and etc. *(4)* The ablation study is in section 5. *(5)*Experiments of more direct engagement with contemporary research on LLM-based generation of high-performance code in Appendix A.6.4. *(6)*Comparison with human expert implementations is in Appendix A.19.

**Evaluation metrics.** *(1)* To evaluate the correctness of generated code, the $pass@k$ metric (Chen et al., 2021) is employed, which calculates the percentage of matrices for which SpMV computation is correct using $k$ (beam search) synthetically generated program samples per matrix. Besides, we also use Compilation Rate (CR) (Kulal et al., 2019) under $k = 1000$ (Appendix A.6) that shows the success rate of compilation among generated code. *(2)* The execution efficiency of the generated SpMV CUDA code is compared with on test matrices using GFLOPS ($10^9$-Giga Floating Point Operations Per Second, higher is better). The platform is the NVIDIA Tesla V100 and A100, based on the Volta and Ampere architecture. Single-precision is used and the CUDA environment is version 12.1. *(3)* The human readability test is in Appendix A.20.

**Datasets.** *Firstly*, we collect the 1,100 matrices with the highest number of non-zero elements from the SuiteSparse Matrix Collection (Davis & Hu, 2011). These matrices span various application domains, and represent a diverse range of sparsity patterns. We divide these matrices into a training set of 700 matrices and a testing set of 400 matrices by the number of non-zero elements. *Secondly*, we conduct further evaluations on the DLMC (Deep Learning Matrix Collection) (Gale et al., 2020; 2019), which includes sparse matrices used in deep learning. When the number of non-zero elements is excessively large, we apply sampling to the input non-zero elements. The details of the sparse matrix dataset are in Appendix A.21. The results of DLMC are elaborated in Appendix A.6.5. The experiments of scalability with sparse matrix features are in Appendix A.14.

### 4.2 RESULTS ON SPMV

**Compilation rate and correct functionality.** Table 1 shows the results of three tasks (including extensions on SpMM in Section 4.3) on 400 test matrices, where the SparseRL with the CodeT5 model outperforms many pretrained LMs of much larger sizes. Despite being applied to the SpMV task, the inherent complexity of the CUDA code results in the generated code having suboptimal quality. The improvement is due to our fine-tuning for specific SpMV CUDA code generation. Specifically, our method leads with a pass@1000 of 49.25 and a CR of 57.50 on SpMV. CR is higher than pass@1000 because a code might compile successfully (high CR), but still have logical errors, thus failing to achieve correct functionality (lower pass@1000). Experiments of more baselines are in Appendix A.6. Pass@1000 is limited by our experimental environment, so the results are marked by '-'. Besides, we observe that although the closed-source model generates code with higher accuracy than our method, there is still a gap in performance (Appendix A.6.2 and Table 7).

**Performance on SuiteSparse Matrix Collection.** The CodeRL and PPOCoder (based on CodeT5-770M under $k = 5000$ on correct programs of partial matrices) are selected to compare the performance with SparseRL (based on Qwen2.5-14B) and the artificial libraries in Figure 3, because the performance results of CodeRL and PPOCoder are better than those of other foundation models (such as Llama/Qwen). The achieved performance is represented in a bin box plot, where the horizontal axis denotes the number of non-zero elements in matrices, and the vertical axis measures performance in GFLOPS. The results demonstrate that SparseRL outperforms other methods across the majority of the matrices. We can observe that: *(1)* On average, SparseRL achieves performance improvements of $3.27\times$, $3.42\times$ on V100 ($3.50\times$, $3.29\times$ on A100) over CodeRL, PPOCoder. *(2)* Notably, on average, SparseRL achieves performance improvements of $1.42\times$, $1.82\times$ on V100 ($1.44\times$, $1.86\times$ on A100) over cuSPARSE, TVM-S. In addition, further performance analysis is elaborated in Appendix A.11.

Table 1: Correct functionality ($pass@k$) and Compilation Rates (CR) under $k = 1000$.

| Model | Size | SpMV | | | | SpMM (col=8) | | | | SpMM (col=32) | | | |
|---|---|---|---|---|---|---|---|---|---|---|---|---|---|
| | | $pass@1$ | $pass@5$ | $pass@1000$ | CR | $pass@1$ | $pass@5$ | $pass@1000$ | CR | $pass@1$ | $pass@5$ | $pass@1000$ | CR |
| Qwen3 | 8B | 8.00 | 12.50 | – | – | 6.75 | 12.00 | – | – | 7.00 | 10.75 | – | – |
| DeepSeek-R1 | 671B | 15.00 | 22.50 | – | – | 16.50 | 20.50 | – | – | 15.50 | 22.75 | – | – |
| DeepSeek-R1-Distill-Qwen | 7B | 9.50 | 16.50 | – | – | 10.00 | 16.75 | – | – | 10.50 | 16.25 | – | – |
| CodeT5 | 770M | 4.75 | 7.50 | 30.25 | 38.00 | 1.75 | 2.25 | 28.00 | 36.00 | 2.00 | 2.25 | 30.00 | 36.25 |
| CodeRL+CodeT5 | 770M | 5.25 | 8.50 | 36.50 | 39.50 | 2.50 | 5.25 | 30.50 | 40.50 | 2.50 | 5.50 | 32.50 | 39.75 |
| PPOCoder+CodeT5 | 770M | 5.75 | 10.00 | 35.50 | 40.75 | 3.50 | 5.00 | 35.50 | 45.50 | 3.50 | 4.75 | 36.00 | 46.75 |
| GPT-o3-pro | - | 25.25 | 32.75 | - | - | 24.25 | 31.50 | - | - | 25.00 | 32.00 | - | - |
| GPT-o4-mini | - | 23.50 | 30.00 | - | - | 22.00 | 28.25 | - | - | 21.25 | 27.25 | - | - |
| GPT-5 | - | 27.00 | 36.50 | - | - | 29.75 | 32.25 | - | - | 26.50 | 31.75 | - | - |
| Claude-sonnet-4 | - | 28.25 | 36.75 | - | - | 24.25 | 31.50 | - | - | 26.00 | 31.50 | - | - |
| SparseRL+CodeT5 | 770M | 9.25 | 15.75 | 48.75 | 56.50 | 9.00 | 15.00 | 45.25 | 56.75 | 8.25 | 14.75 | 47.00 | 57.25 |
| SparseRL+Qwen2.5 | 7B | 9.75 | 16.00 | 48.75 | 57.00 | 10.00 | 15.00 | 46.50 | 57.00 | 8.50 | 15.00 | 46.75 | 58.00 |
| SparseRL+Qwen2.5 | 14B | 10.25 | 16.50 | 49.25 | 57.50 | 10.25 | 16.00 | 47.50 | 58.75 | 9.25 | 15.25 | 47.75 | 59.00 |
| SparseRL+Qwen3 | 14B | 16.25 | 21.00 | – | – | 18.25 | 21.00 | – | – | 18.25 | 22.25 | – | – |
| SparseRL+GLM-Z1 | 9B | 15.50 | 19.75 | – | – | 16.50 | 20.00 | – | – | 16.00 | 21.00 | – | – |
| SparseRL+DeepSeek-Coder | 6.7B | 17.75 | 20.25 | – | – | 17.50 | 21.00 | – | – | 18.75 | 22.25 | – | – |

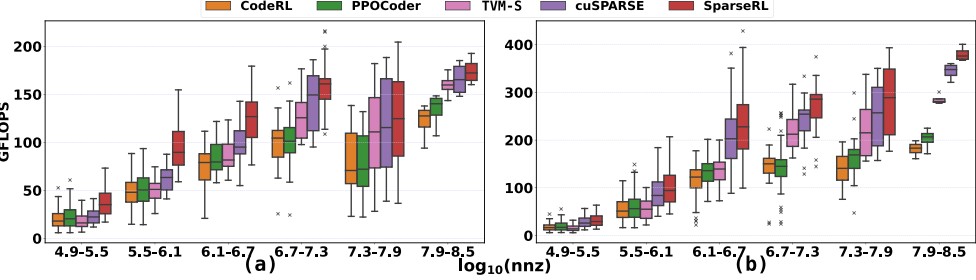

Figure 3: Performance (GFLOPs) of SpMV across SuiteSparse Matrices on (a) V100 (b) A100 GPUs.

## 4.3 EXTENSIONS ON SPMM

We extend our method to another sparse CUDA code task of sparse matrix-dense matrix multiplication (SpMM). SpMM is another crucial operation in high-performance computing, which has been applied to GNN (Wang et al., 2023b; Huang et al., 2020) and pruning LLMs (Xia et al., 2023; Fan et al., 2024). It multiplies an $M \times K$ sparse matrix A and a $K \times N$ dense matrix B to output an $M \times N$ dense matrix C (i.e., C = AB). Similar to SpMV, implementing high-performance SpMM on GPUs is challenging due to the complex nature of sparse matrices. The platform is the NVIDIA Tesla A100 and single-precision is used. Execution efficiency is compared using TFLOPS ($10^{12}$-Tera Floating Point Operations Per Second). In addition, we adjust the number of columns of the dense input matrix $B$, as it also affects execution time. The **baseline** for comparison is the same as SpMV. The differences of performance optimization characteristics between SpMV and SpMM are in Appendix A.13.

**Compilation rate, correct functionality and achieved performance on SuiteSparse matrix**. Table 1 shows that the SparseRL with the CodeT5 model can achieve significant performance gains on SpMM tasks. Figure 4 illustrates the performance of partial matrices across various methods. The achieved performance is represented in a bin box plot, where the horizontal axis denotes the number of non-zero elements in the matrices, and the vertical axis measures performance in TFLOPS. The results demonstrate that SparseRL consistently outperforms other methods across the majority of the matrices analyzed. We can observe that: **(1)** Notably, SparseRL achieves the average speedup of $6.80/4.50\times$ over CodeRL at 8/32 column, $6.\overline{60}/4.88\times$ on A100 over PPOCoder at 8/32 column. **(2)** SparseRL achieves performance improvements of average $2.32/1.22\times$ on A100 over Sputnik at $8/\overline{32}$ column, $6.39/4.38\times$ over cuSPARSE at 8/32 column.

## 5 ABLATION STUDY

### 5.1 ABLATION STUDY OF PRE-TRAINING/SFT/RL PHASES

Table 2 presents a direct ablation comparison between the three phases (Pre-training/ SFT/ RL) of SparseRL. The results clearly show that (1) Pretaining phase is critical for generating valid CUDA code, where CUDA-specific pre-training equips the model with parallel computing and GPU memory

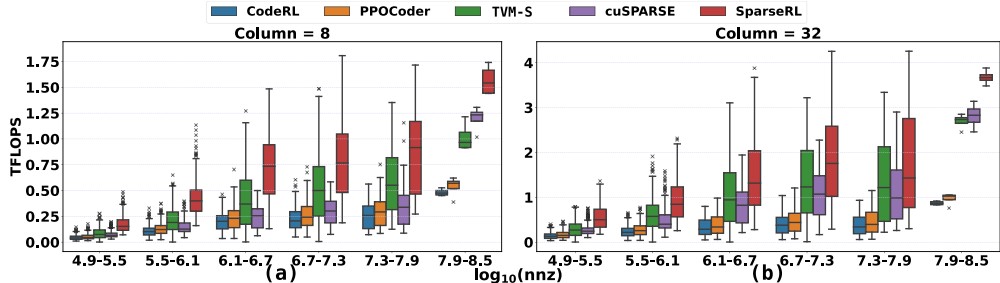

Figure 4: Performance (TFLOPs) of SpMM across SuiteSparse Matrices with (a) column = 8 (b) column = 32 on A100 GPUs. Column is the number of input matrix $B$ columns.

Table 2: Ablation comparison between the three phases (Pre-training/SFT/RL).

| Metric | Pretrain | | | No Pretrain | | |
|---|---|---|---|---|---|---|
| | SFT | PPO | SFT + PPO | SFT | PPO | SFT + PPO |
| SpMV pass@1000 | 32.75 | 15.25 | 49.25 | 25.50 | 12.00 | 40.75 |
| SpMV Compilation Rate | 41.25 | 22.50 | 57.50 | 37.25 | 13.25 | 53.50 |
| Average Execution Speed (GFLOPS) | 89.25 | 50.36 | 116.20 | 70.32 | 30.83 | 95.22 |

management knowledge. (2) SFT phase is the fundamental stage of post training for the entire model. Especially in the SFT stage, we performed modality transformation (input sparse matrix, output code), which is important for the subsequent output of the model. If we only do the RL stage, it is difficult to obtain the correct code, and even more difficult to obtain high-performance correct code. (3) RL phase, driven by the hierarchical reward function (incorporating efficiency and memory constraints), is also important and significantly enhances both correctness and performance beyond the SFT baseline. The three phases together contributed to the highest performance and accuracy of the code, indicating that both stages are necessary.

## 5.2 ABLATION STUDY OF SPARSE MATRIX EMBEDDING

In this ablation study, we focus on understanding how these different sparse matrix embedding strategies affect the performance of SparseRL in terms of correct functionality and compilation rates for SpMV and SpMM tasks. Table 3 presents the results of this study for models with different embedding techniques under $k = 1000$. The *Raw* approach simply feeds the original numerical values of the indices into the model, without any transformation. The *Max-Min* normalization technique scales the values of the indices to the range between 0 and 1, using the maximum and minimum values within the dataset for normalization. These results clearly demonstrate that the sinusoidal embedding is the most effective among the three strategies for enabling the model to generate high-quality code for sparse matrix operations. More experiments of sparse matrix embedding are in Appendix A.8.

Table 3: Ablation study of sparse matrix embedding on correct functionality ($pass@k$) and Compilation Rates (CR) under $k = 1000$.

| Sparse Matrix Embedding | Model | Size | SpMV | | | | SpMM (col=8) | | | |
|---|---|---|---|---|---|---|---|---|---|---|
| | | | $pass@1$ | $pass@5$ | $pass@1000$ | CR | $pass@1$ | $pass@5$ | $pass@1000$ | CR |
| Raw | SparseRL+CodeT5 | 770M | 5.25 | 10.50 | 40.75 | 49.50 | 5.00 | 12.00 | 38.50 | 48.75 |
| Max-Min | SparseRL+CodeT5 | 770M | 8.25 | 14.00 | 43.25 | 51.75 | 7.00 | 13.50 | 40.00 | 52.75 |
| Sinusoidal | SparseRL+CodeT5 | 770M | 9.25 | 15.75 | 48.75 | 56.50 | 9.00 | 15.00 | 45.25 | 56.75 |

## 5.3 ABLATION STUDY OF REWARD COMPONENTS

We aim to understand the impact of different components in our SparseRL method on the performance of SpMV. Figure 5 presents the results of our ablation studies on two different GPU architectures. We compare four approaches: (i) cuSPARSE, which is the NVIDIA library for sparse matrix operations; (ii) *SparseRL (base)* means the supervised fine-tuning (SFT); (iii) *Op1* represents the utilization of RL with only the correctness reward; (iv) *Op2* represents the addition of execution efficiency reward.

Figure 5: Ablation Studies on (a) V100 and (b) A100.

In Figure 5, we observe that the basic *SparseRL (base)* achieves similar performance with cuSPARSE in most cases. When we add the RL method to form *SparseRL (base+op1)*, the performance further improves for matrices like water_tank and helm2d03, showing the positive impact of this specific optimization. Moreover, the *SparseRL (base+op1+op2)* variant achieves the highest GFLOPS in several cases, such as nemeth22 and ga2010, demonstrating that combining execution efficiency reward can lead to even better performance. Overall, these ablation study results clearly show that each component in our SparseRL method contributes to improving the performance.

## 5.4    ABLATION STUDY OF RL ALGORITHM CHOICES

We actually tried PPO, GRPO, and Reinforce++, but found that PPO's performance was already good enough, so we chose PPO. For specific details, please refer to the following comparison: (1) GRPO's gradient regulation mechanism reduces training instability but increases computational overhead by 12% (training time extends from 5 to 5.6 days on 8 GPUs). (2) GRPO achieves 48.9 pass@1000 and 113.5 GFLOPS, which are 0.7% and 2.3% lower than PPO, respectively. (3) The innovation of our method primarily lies in the process, and our approach is robust to the selection of reinforcement learning algorithms

The experimental results on the same training/test split are: (1) GRPO: Achieves similar correctness (pass@1000: 48.9 vs. SparseRL's 49.25 on SpMV) but has 12% higher training overhead due to more complex gradient regulation. Execution speed is 2.1% lower than SparseRL. (Due to the randomness of reinforcement learning itself, the test results may fluctuate, and GRPO also occasionally surpasses PPO). (2) Reinforce++: Shows 3.5% lower correctness (pass@1000: 47.6) and 5.3% slower execution speed compared to SparseRL on SpMV. This is because Reinforce++ lacks PPO's clipped objective, leading to unstable training when balancing code correctness and efficiency.

Additionally, for our framework, PPO can be replaced with other state-of-the-art reinforcement learning algorithms, which is only a part of the pipeline in our method. As demonstrated in the experiment, although there may be fluctuations in performance, it does not affect our performance improvement.

## 6    CONCLUSION

**Discussions**. While SparseRL is currently optimized for sparse matrix operations, its core design can be extended to other general code optimization tasks. We outline the extension potential of our framework: (1) Input representation: Replace sparse matrix indices with task-specific structural features (add some other "multi-model adapters" to capture information from non-matrix input. For example, use GNN to extract feature from program dependency graph or use UniXcoder (Guo et al., 2022)/GraphCodeBERT (Guo et al., 2021) as code embedding techniques). (2) Reward function: Adapt the hierarchical reward to task-specific metrics (e.g., loop execution time, parallelization speedup, memory bandwidth utilization). (3) RL pipeline: Reuse the pretrain→SFT→RL workflow with task-specific training data.

**Limitations.** Despite the promising results, SparseRL has several failure cases (Appendix A.15) and limitations. First, the RL-based optimization process is computationally expensive during fine-tuning (Appendix A.16), because of the interaction with the compiler and executor to obtain correctness and execution time feedback. Second, since both the time required to generate Sparse CUDA code and the execution time contribute to the overall overhead in real-world applications, our method is particularly well-suited for scenarios where sparse code can be reused repeatedly (Appendix A.17). Third, the extension to other hardware-backends is elaborated in Appendix A.18.

**Conclusion.** We propose the SparseRL method to address the challenges in generating high-performance CUDA code and apply it to sparse matrix operation tasks. By integrating deep reinforcement learning and pre-trained models, SparseRL has demonstrated significant improvements in both the correctness and execution efficiency of generated sparse CUDA code.

ACKNOWLEDGMENTS

The authors sincerely thank anonymous reviewers for their insightful suggestions. The work is supported by National Natural Science Foundation of China, under Grant No. T2125013, 62032023, 62172391.

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

# A  APPENDIX

## A.1  THE KEY BOTTLENECKS PREVENTING THE MODEL FROM FULLY SOLVING THIS PROBLEM

The works (Ouyang et al., 2025; Won et al., 2023) demonstrated the sparse high-performance parallel tasks are difficult, compared with the code implementation (python/C++) of algorithm problems (Le et al., 2022) and dense computation (Zheng et al., 2020). Here, we give several reasons to illustrate the bottlenecks of our work from GPU kernel and sparse computation aspect.

From GPU kernel aspect, (1) LLMs (base models) face the problem of insufficient training data volume, which restricts their performance in GPU kernel generation. CUDA is a low-resource language in open-source training data (about 0.073% (Ouyang et al., 2025)). (2) LLMs fail to effectively utilize hardware resources (e.g., shared memory / tensor core), resulting in low-efficiency generated kernels, which we have partially solved by SFT of high-performance CUDA code and RL of memory-limit reward.

From sparse computation aspect, (1) The parallel implementation of sparse operations is inherently more difficult than dense operations, because it is necessary to consider that the GPU units have to process different numbers of non-zero elements during the parallel computation and then accumulate the results, which is more likely to lead to calculation errors in the final result. (2) Given the complexity of sparsity patterns and the intricate relationship between sparse matrix features, GPU architecture characteristics, and performance, it is impossible to develop a one-size-fits-all code-implementation for sparse computation (Won et al., 2023).

## A.2  THEORETICAL FOUNDATION FOR SINUSOIDAL EMBEDDING

The sinusoidal encoder (Vaswani et al., 2017), widely adopted across numerous fields such as audio signal and computer vision (Benammar et al., 2017; Rußwurm et al., 2023), has shown great potential in capturing sequential and positional information.

Following these previous studies, we have some insights into why sinusoidal embeddings are well-suited for sparse matrix representation and the motivation for its applicability.

1. **Positional Invariance and Multi-Scale Representation**
    - Our sinusoidal embeddings (from Eq. 3 in Sec3) treat row and column indices like continuous waves:

$$\text{PE}(ind, 2j) = \sin\left(\frac{ind}{10000^{2j/d_{\text{model}}}}\right), \quad \text{PE}(ind, 2j+1) = \cos\left(\frac{ind}{10000^{2j/d_{\text{model}}}}\right)$$

    - The base value 10,000 combined with the exponent $2j/d_{\text{model}}$ creates wavelengths that grow geometrically. Dimensions at the lower end (small $j$) capture broad positional patterns—think block-level structures. Meanwhile, higher dimensions (large $j$) pick up fine details like local clusters of non-zero entries.

- This multi-scale approach handles **irregular sparsity patterns** (like power-law distributions in GNNs) far better than fixed-interval positional encoding ever could.

2. **Relative Positional Awareness**

- In sparse matrices, how non-zero elements are *relatively positioned* drives memory access patterns (coalesced vs. scattered loads). Sinusoidal embeddings shine here because shifting positions (Li et al., 2021) follows a neat pattern:

$$\text{PE}(x) = [\sin(\omega_1 x), \cos(\omega_1 x), \dots, \sin(\omega_k x), \cos(\omega_k x)]$$

For any shift $\delta$, we have:

$$\text{PE}(x + \delta) = [\sin(\omega_1 x + \omega_1 \delta), \cos(\omega_1 x + \omega_1 \delta), \dots]$$

Using angle addition identities:

$$\sin(\omega x + \omega \delta) = \sin(\omega x)\cos(\omega \delta) + \cos(\omega x)\sin(\omega \delta)$$

So:

$$\text{PE}(x + \delta) = A(\delta) \cdot \text{PE}(x)$$

Where $A(\delta)$ is a **linear transformation matrix** that rotates each frequency component by $\omega \delta$.

**Key Insight**: A linear model using sinusoidal PE can *linearly model shifts*. This allows the system to reason about relative differences. Essentially, models can generalize across matrices with similar non-zero distributions even if their coordinates are shifted.

3. **Sparsity-Induced Sparsity in Embeddings**

- Real-world sparse matrices often show **fractal** or **hierarchical** structures (scale-free networks are a classic example). Sinusoidal encodings preserve local proximity: indices (r, c) that are close yield embedding vectors with high cosine similarity, making them suitable for learning locality-aware behaviors in sparse patterns:
  - They smoothly map huge index ranges (e.g., millions of rows) into $[-1, 1]$ without losing information—unlike crude scaling methods.
  - They preserve neighborhood relationships: Elements with similar $(r_i, c_i)$ indices get similar embeddings. This directly helps our RL policy generate CUDA kernels optimized for local memory access patterns.

**Why Alternatives Fail**

- Raw Indices: Scale poorly for large matrices (e.g., $r_i = 10^6$ vs. $r_j = 1$), causing numerical instability and poor generalization.

- Max-Min Normalization: Loses relative positional relationships; e.g., $(r_i, c_i) = (1, 1)$ and $(2, 2)$ may map to distant values after scaling.

- Learnable Embeddings: Require fixed vocabulary sizes (infeasible for arbitrary indices) and fail to extrapolate to unseen matrix dimensions.

A.3    CODE CORRECTNESS VERIFICATION

For stopping the generation of error code early during the generation process and reducing unnecessary computations, we integrate a dynamic syntax correctness verification mechanism during the decoding process. Specifically, for the partially generated CUDA code at the step when a complete line of code is generated (marked by the token of ";"), we parse its intermediate code to construct a syntax tree and maintain a symbol table to record variable names, data types, and scoping rules. If the partial program violates CUDA syntax rules (e.g., unmatched parentheses, undeclared variables, or incorrect kernel launch syntax) or introduces semantic conflicts (e.g., variable name reuse with incompatible types), the generation process is terminated early. The code extraction tool is modified from the public QwenLM repository.

## A.4   REWARD DETAILS

### A.4.1   SENSITIVITY STUDY OF $r_{\text{EFF}}$ AND $r_{\text{PENALTY}}$ IN REWARD HYPERPARAMETERS

- **Efficiency scaling factor ($r_{\text{eff}}$)**: Fixed at **1.0**. This value balances the magnitude of efficiency rewards with correctness rewards (Eq. 8), ensuring the policy prioritizes both code correctness and execution speed without bias.

- **Memory penalty ($r_{\text{penalty}}$)**: Set to **0.3**. This penalty discourages excessive memory use (e.g., shared memory $>$48 KB on V100) while preserving positive rewards for correct, efficient code—for example, a kernel with successful compilation (+0.5) and correct execution (+0.5) still yields a net positive reward ($0.5 + 0.5 - 0.3 = 0.7$) even with memory overuse, avoiding dis-incentivizing valid code structures.

We evaluated SpMV performance (pass@1000, Compilation Rate (CR), average GFLOPS on V100) by varying $r_{\text{penalty}}$ (0.1, 0.3, 0.5) on 400 test matrices:

- $r_{\text{eff}} = 0.5$: Underscales efficiency rewards, leading to lower speed (102.3 GFLOPS) despite modest correctness.

- $r_{\text{eff}} = 1.5$: Overscales efficiency rewards, marginally boosting speed (118.7 GFLOPS) but reducing pass@1000 (48.50 vs. 49.25 for base), as the policy prioritizes speed over correctness.

Table 4: Impact of Varying $r_{\text{eff}}$ (Fixed $r_{\text{penalty}} = 0.3$)

| $r_{\text{eff}}$ | SpMV pass@1000 | SpMV CR (%) | Avg. GFLOPS |
|---|---|---|---|
| 0.5 | 45.75 | 55.20 | 102.3 |
| 1.0 (Base) | 49.25 | 57.50 | 116.2 |
| 1.5 | 48.50 | 56.80 | 118.7 |

Varying $r_{\text{penalty}}$ (0.1, 0.3, 0.5) on 400 test matrices:

- $r_{\text{penalty}} = 0.1$: Weak penalty fails to curb excessive memory use, leading to lower speed (108.5 GFLOPS) due to suboptimal memory allocation.

- $r_{\text{penalty}} = 0.5$: Overpenalization reduces pass@1000 (46.80) and CR (54.30), as the policy avoids valid memory-intensive optimizations (e.g., shared memory for cache locality).

Table 5: Impact of Varying $r_{\text{penalty}}$ (Fixed $r_{\text{eff}} = 1.0$)

| $r_{\text{penalty}}$ | SpMV pass@1000 | SpMV CR (%) | Avg. GFLOPS |
|---|---|---|---|
| 0.1 | 47.25 | 58.10 | 108.5 |
| 0.3 (Base) | 49.25 | 57.50 | 116.2 |
| 0.5 | 46.80 | 54.30 | 119.1 |

These results confirm the base values ($r_{\text{eff}} = 1.0$, $r_{\text{penalty}} = 0.3$) are optimal—balancing correctness, compilation success, and execution speed while ensuring PPO stability and generalizability to unseen matrices.

### A.4.2   $R_{efficiency}$ NORMALIZATION IN REWARD

To address runtime variability across matrices (e.g., small matrices with $t_{base} = 1$ ms vs. large matrices with $t_{base} = 100$ ms), we apply \*\*per-matrix z-score normalization\*\* to the efficiency reward ($R_{efficiency}$, Eq. 9):

$$R_{efficiency}^{\text{norm}} = r_{eff} \times \frac{\frac{t_{base}(X)}{t(\hat{Y}, X)} - \mu_X}{\sigma_X} \times \mathbb{I}_{test}$$

Here, $\mu_X$ and $\sigma_X$ are the mean and standard deviation of the speedup ratio ($\frac{t_{base}(X)}{t(\hat{Y},X)}$ vs. cuSPARSE) across 1000 validation matrices. This restricts $R_{efficiency}^{norm}$ to $[-2, 2]$ for 95% of matrices, ensuring rewards are comparable regardless of a matrix's inherent size or sparsity. In most of the time, this ratio is bounded. If not, we do truncation to make it within the range (restricted within $[-2, 2]$).

### A.4.3 WEIGHT ADJUSTMENT FOR CORRECTNESS AND EFFICIENCY

Correctness reward and the efficiency reward are not equally weighted in the final reward. To address this question, we have conducted experiments with adjustable weight coefficients:

$$R_{final} = \alpha \cdot R_{correctness} + (1 - \alpha) \cdot R_{efficiency} - r_{penalty} \cdot \mathbb{I}_{memory}$$

where $\alpha \in [0, 1]$ controls the trade-off between correctness and efficiency. Experimental results on the SuiteSparse test set:

- $\alpha = 0.9$ (prioritize correctness): pass@1000 = 51.2 ($\uparrow$4.1%), but execution speed = 98.7 GFLOPS ($\downarrow$15.1%) compared to the original $\alpha = 0.5$.
- $\alpha = 0.5$ (balanced): pass@1000 = 49.25, execution speed = 116.2 GFLOPS (optimal trade-off).
- $\alpha = 0.1$ (prioritize efficiency): execution speed = 123.5 GFLOPS ($\uparrow$6.3%), but pass@1000 = 38.7 ($\downarrow$21.4%).

We confirm that the original weight ($\alpha = 0.5$) achieves the good balance between correctness and efficiency, which is critical for high-performance code generation.

### A.5 PPO TRAINING DIAGNOSTICS (KL DIVERGENCE, ENTROPY, VALUE LOSS, REWARD)

Figure 6 provides a comprehensive view of PPO training stability across 700 epochs, with key observations:

- **Value Loss (Subplot (a))**: Both train and test value loss decline steadily from 0.7 to near 0.0 by epoch 700. This confirms the critic network ($V_\phi(s_t)$) accurately estimates the final reward ($R_{final}$), enabling reliable advantage calculation (GAE, §3.3).
- **Entropy (Subplot (b))**: Policy entropy decreases from 1.0 to 0.15, indicating the policy converges to consistent, high-reward code patterns. The gradual decline also shows the policy retains enough exploration to adapt to diverse sparse matrices (e.g., irregular vs. block-dense structures).
- **KL Divergence (Subplot (c))**: KL divergence between old and new policies stabilizes within 0.1–0.3 after epoch 100. This ensures policy updates are incremental, avoiding catastrophic shifts that could harm performance.
- **Reward (Subplot (d))**: The average reward rises from -0.5 to 1.5, reflecting the policy's ability to learn increasingly correct and efficient code generation—consistent with the 30% speedup over baselines reported in §4.2.

These diagnostics collectively validate that PPO training is stable, with the policy learning to balance correctness, efficiency, and memory constraints without overfitting or instability.

### A.6 MORE BASELINE MODELS AND OBSERVATIONS

### A.6.1 OPEN-SOURCE MODELS

We add more experimental comparisons on more recent publicly available LLMs in Table 6. For the dilemma of general-purpose LLMs, DeepSeek-R1 (671B) achieved the highest accuracy (pass@5) (22.75%), but its performance plummeted by 27.5%.

From the results, we can conclude that larger models excel at syntactic correctness but lack hardware optimization knowledge. Qwen3-32B's accuracy (19.00%) is close to SparseRL+Qwen3-14B (22.25%), but its performance lags by 34.5%. General-purpose code generation cannot replace domain-specific optimization.

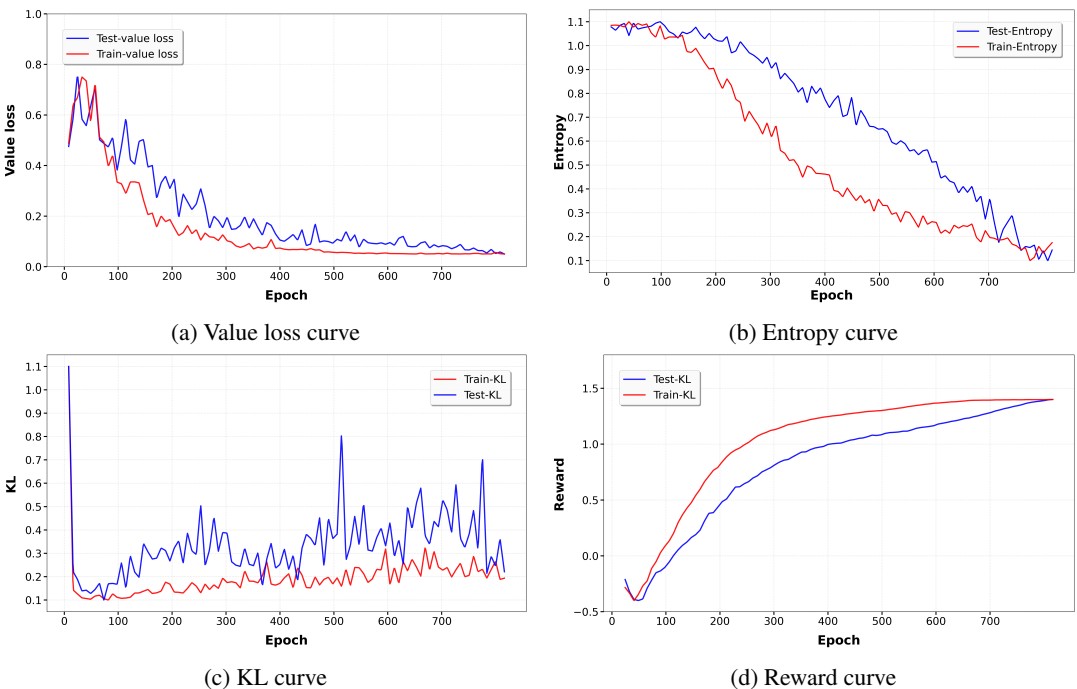

(a) Value loss curve

(b) Entropy curve

(c) KL curve

(d) Reward curve

Figure 6: PPO diagnostics (KL, entropy, value loss, reward) over training.

Additionally, we find that the distilled version of DeepSeek-R1 (7B/14B) achieved accuracy close to SparseRL (17.50% vs. 16.50%), but its performance lags by over 30%. We infer that distillation transfers semantic knowledge but cannot transfer hardware optimization strategies.

Table 6: Correct functionality ($pass@k$) and Performance (Speedup vs. cuSPARSE) comparison for correct generated program on $pass@5$.

| Model | Size | SpMV | | | SpMM (col=8) | | | SpMM (col=32) | | |
|---|---|---|---|---|---|---|---|---|---|---|
| | | $pass@1$ | $pass@5$ | Perf. | $pass@1$ | $pass@5$ | Perf. | $pass@1$ | $pass@5$ | Perf. |
| DeepSeek-R1 | 671B | 15.00 | 22.50 | **-27.5%** | 16.50 | 20.50 | **-27.8%** | 15.50 | 22.75 | **-27.2%** |
| DeepSeek-R1-Distill -Qwen-7B | 7B | 9.50 | 16.50 | **-31.1%** | 10.00 | 16.75 | **-32.7%** | 10.50 | 16.25 | **-30.9%** |
| DeepSeek-R1-Distill -Qwen-14B | 14B | 10.50 | 17.50 | **-32.3%** | 11.75 | 17.50 | **-27.9%** | 10.00 | 17.00 | **-28.1%** |
| Qwen3 | 8B | 8.00 | 12.50 | **-36.8%** | 6.75 | 12.00 | **-37.5%** | 7.00 | 10.75 | **-36.2%** |
| Qwen3 | 14B | 8.25 | 13.75 | **-35.9%** | 8.00 | 14.00 | **-36.3%** | 8.50 | 13.50 | **-35.7%** |
| Qwen3 | 32B | 10.25 | 18.50 | **-36.1%** | 11.50 | 20.00 | **-35.8%** | 12.00 | 19.00 | **-34.5%** |
| SparseRL+Qwen2.5 | 7B | 9.75 | 16.00 | **+27.5%** | 10.00 | 15.00 | **+39.1%** | 8.50 | 15.00 | **+15.8%** |
| SparseRL+Qwen2.5 | 14B | 10.25 | 16.50 | **+30.2%** | 10.25 | 16.00 | **+42.7%** | 9.25 | 15.25 | **+17.3%** |
| SparseRL+Qwen3 | 8B | 15.75 | 20.00 | **+32.8%** | 15.00 | 22.00 | **+42.5%** | 15.50 | 21.00 | **+20.1%** |
| SparseRL+Qwen3 | 14B | 16.25 | 21.00 | **+34.5%** | 18.25 | 21.00 | **+47.2%** | 18.25 | 22.25 | **+21.9%** |

### A.6.2 CLOSE-SOURCE MODELS

We add more experimental comparisons on closed-source models with API calls (Claude-3.7/4.0, GPT-4.1, o3-pro and DeepSeek-R1-0528) in Table 7.

**Core Finding: Correctness ≠ Performance Optimization.** From the experiment, we observe that although the closed-source model generates code with higher accuracy than our method, there is still a gap in performance. The comparison also shows that the performance of the generated programs does not vary much between closed-source models. Closed-source models excel at syntactic correctness but lack hardware optimization knowledge.

The reasons are from two aspects. (1) We observe that the programs generated by SparseRL make use of more complex optimization techniques and instructions in CUDA, such as the use of shared memory and warp shuffle instructions. These complex program statements lead to a decrease in accuracy. (2) Our method uses RL to fine-tune the performance of each test matrix in training, so that the model can perceive and generate different programs for different sparse matrices, while the programs output by closed-source models for different sparse matrices are often similar.

Table 7: Correct functionality ($pass@k$) and performance (Speedup vs. cuSPARSE) comparison for correct generated program on $pass@5$.

| Model | Size | SpMV | | | SpMM (col=8) | | | SpMM (col=32) | | |
|---|---|---|---|---|---|---|---|---|---|---|
| | | $pass@1$ | $pass@5$ | Perf. | $pass@1$ | $pass@5$ | Perf. | $pass@1$ | $pass@5$ | Perf. |
| Claude-3.7 | - | 18.50 | 24.25 | -30.8% | 17.00 | 22.00 | -30.2% | 17.25 | 23.50 | -30.7% |
| Claude-4.0 | - | 21.00 | 27.50 | -27.1% | 19.75 | 25.25 | -26.3% | 20.00 | 26.75 | -25.9% |
| GPT-4.1 | - | 23.50 | 30.00 | -25.7% | 22.25 | 28.75 | -26.9% | 23.00 | 29.25 | -24.8% |
| o3-pro | - | 25.25 | 32.75 | -18.2% | 24.25 | 31.50 | -19.1% | 25.00 | 32.00 | -17.5% |
| DeepSeek-R1-0528 | - | 26.75 | 34.50 | -17.3% | 25.75 | 33.25 | -19.7% | 26.25 | 34.25 | -17.9% |
| GPT-o3-pro | - | 25.25 | 32.75 | -18.2% | 24.25 | 31.50 | -19.1% | 25.00 | 32.00 | -17.5% |
| GPT-o4-mini | - | 23.50 | 30.00 | -23.7% | 22.00 | 28.25 | -23.8% | 21.25 | 27.25 | -22.8% |
| GPT-5 | - | 27.00 | 36.50 | -18.0% | 29.75 | 32.25 | -18.3% | 26.50 | 31.75 | -20.5% |
| Claude-sonnet-4 | - | 28.25 | 36.75 | -19.2% | 24.25 | 31.50 | -18.0% | 26.00 | 31.50 | -18.5% |
| SparseRL+Qwen2.5 | 7B | 9.75 | 16.00 | **+27.5%** | 10.00 | 15.00 | **+39.1%** | 8.50 | 15.00 | **+15.8%** |
| SparseRL+Qwen2.5 | 14B | 10.25 | 16.50 | **+30.2%** | 10.25 | 16.00 | **+42.7%** | 9.25 | 15.25 | **+17.3%** |
| SparseRL+Qwen3 | 8B | 15.75 | 20.00 | **+32.8%** | 15.00 | 22.00 | **+42.5%** | 15.50 | 21.00 | **+20.1%** |
| SparseRL+Qwen3 | 14B | 16.25 | 21.00 | **+34.5%** | 18.25 | 21.00 | **+47.2%** | 18.25 | 22.25 | **+21.9%** |

### A.6.3 APPROPRIATENESS OF CODERL/PPOCODER

CodeRL and PPOCoder are appropriate baselines because:

- They are representative RL-based code generation methods, sharing the same core paradigm (pretrained LLM + RL) as SparseRL.
- While not explicitly trained for GPU kernel optimization, they are the most relevant baselines for evaluating RL-driven code generation performance.

### A.6.4 DIRECT ENGAGEMENT WITH CONTEMPORARY RESEARCH ON LLM-BASED GENERATION OF HIGH-PERFORMANCE CODE

We elaborate on how each key innovation directly addresses the unique challenges of sparse computation by the direct engagement with two contemporary research (PerfCodeGen (Peng et al., 2025) and LLM4EFFI (Ye et al., 2025)) on LLM-based generation of high-performance code, with concrete links to experimental results:

**PerfCodeGen** (Peng et al., 2025) enhances general code efficiency through iterative self-refinement, using execution feedback from test cases, specifically identifying the most time-consuming unit tests to guide optimizations. The superiority of SparseRL stems directly from how its design addresses the unique challenges of sparse GPU computation:

- **Domain specialization**: Unlike PerfCodeGen's focus on general code (e.g., standard algorithms in HumanEval), SparseRL targets sparse matrix operations (SpMV/SpMM) on GPUs. These operations demand mastery of irregular memory access patterns and GPU parallelism areas, where PerfCodeGen's general feedback loops fall short. For example, SparseRL learns to align non-zero elements with GPU memory banks to avoid bank conflicts, a optimization irrelevant to PerfCodeGen's target tasks but critical for reducing sparse computation runtime.
- **Feedback mechanism**: PerfCodeGen relies on verbalized feedback (e.g., "this test case is slow"), which struggles to capture low-level hardware nuances. In contrast, SparseRL's hierarchical RL reward (incorporating compilation success, correctness, execution time, and memory usage) directly incentivizes hardware-aware tweaks. For instance, the efficiency reward (tied to cuSPARSE runtime) drives the model to discover warp-padding strategies

Table 8: Results of correct functionality (pass@k) and performance evaluated by speedup. Correctly generated code on pass@1 is used to evaluate the speedup compared with cuSPARSE.

| Method | Size | pass@1 | Speedup (vs. cuSPARSE) |
|---|---|---|---|
| LLM4EFFI + Qwen2.5-Coder (mentioned in their paper) | 32B | 10.00 | -25.1% |
| LLM4EFFI + Qwen3 (We add additionally) | 14B | 11.00 | -15.5% |
| SparseRL + Qwen3 | 14B | **16.00** | +32.5% |

for unbalanced rows, a key reason SparseRL outperforms PerfCodeGen-derived approaches on sparse tasks.

- **Input representation**: PerfCodeGen treats inputs as natural language descriptions, but SparseRL's sinusoidal embeddings of row/column indices explicitly model sparse matrix structure. This allows it to generate matrix-specific kernels (e.g., different strategies for clustered vs. scattered non-zeros), boosting compilation rates compared to using raw indices (Appendix A.4 Table 5).

**LLM4EFFI** (Ye et al., 2025) optimizes general code via logic-domain algorithm exploration and code-domain implementation, prioritizing efficiency first. SparseRL's performance gains arise from design choices tailored to sparse GPU operations:

- **Scope and hardware awareness**: LLM4EFFI focuses on general code (e.g., mathematical functions) and lacks GPU-specific optimization targets. SparseRL, by contrast, targets CUDA kernels for sparse operations, learning to optimize thread block sizes, shared memory usage, and atomic reductions, which are critical for GPU performance. This specialization explains why SparseRL achieves 1.82x speedup on A100 (Sec. 4.2) over TVM-S (a generic sparse compiler) on SpMV tasks.

- **Learning paradigm**: LLM4EFFI uses supervised fine-tuning with synthetic test cases, which cannot capture GPU runtime metrics. SparseRL's deep RL (PPO) directly optimizes for execution time on GPUs, enabling it to learn non-intuitive tweaks like using __shfl__ instructions for intra-warp communication, and reducing global memory access on matrices with high row overlap.

- **Adaptivity to input structure**: Sparse computation performance is highly input-dependent, but LLM4EFFI generates static code. SparseRL's index embeddings allow it to customize kernels for each matrix's non-zero distribution (e.g., padding short rows for load balance), a capability that improves pass@1000 (in Section 5.2 Table 3) compared to non-embedded approaches.

These design choices directly address the unique challenges of sparse computation, causing SparseRL's superior performance. We are the first to apply RL to sparse computation, enabling these hardware-aware and input-adaptive optimizations that general frameworks cannot replicate.

We provide the experimental results comparing both frameworks on the SpMV task using the SuiteSparse test set (100 matrices) under identical hardware/software conditions in Table 8. The experimental setup is as follows:

- **Task**: Sparse Matrix-Vector multiplication (SpMV)

- **Dataset**: 100 test matrices randomly sampled from SuiteSparse (among 400 test matrices in our paper).

- **Timeline**: Generation time (5 hours; average 3 min for each matrix) and SpMV execution time (3 hours) of LLM4EFFI

- **Baseline**: LLM4EFFI (Ye et al., 2025) using its official implementation on anonymous Github and recommended settings.

- Since LLM4EFFI follows a similar generate-verify-profile workflow, we can adapt LLM4EFFI to accept identical sparse matrix input (row/column indices) but without sinusoidal embeddings (its design lacks this modality).
- Optimization target: SpMV CUDA kernel generation.

- **Evaluation Metrics**:
  - **Correctness**: pass@1.
  - **Performance**: GFLOPS (speedup over cuSPARSE).

### A.6.5 SpMV Performance on DLMC matrix

To broaden the matrix diversity, we conduct additional performance evaluations on the DLMC (Deep Learning Matrix Collection), which includes sparse matrices frequently used in deep learning. *(1)* On average, SparseRL achieves performance improvements of $10.21\times$, $1.28\times$ on V100 ($10.70\times$, $1.33\times$ on A100) over CodeRL, PPOCoder. *(2)* Notably, on average, SparseRL achieves performance improvements of $1.29\times$, $1.69\times$ on V100 ($1.25\times$, $1.66\times$ on A100) over cuSPARSE, TVM-S.

We choose five representative matrices of 0.95 magnitude pruning (Han et al., 2015) based on non-zero elements to illustrate performance in Table 9 (where m1, m2, m3, m4, m5 are *symbol-modality-33288-512-shared-weights-0-aux*, *body-decoder-layer-1-ffn-conv2-fully-connected*, *bottleneck-2-block-group-projection-block-group4*, *final-dense*, *bottleneck-2-block-group4-2-1*).

Table 9: Performance (GFLOPs) Comparison of different methods on DLMC dataset.

| Matrix | CodeRL | PPOCoder | TVM-S | cuSPARSE | SparseRL |
|--------|--------|----------|-------|----------|----------|
| m1 | 27.76 | 27.32 | 96.41 | 97.23 | 121.82 |
| m2 | 20.98 | 25.44 | 67.49 | 58.34 | 88.93 |
| m3 | 17.53 | 21.02 | 48.21 | 48.62 | 64.56 |
| m4 | 10.11 | 15.67 | 26.79 | 31.67 | 43.50 |
| m5 | 15.82 | 12.10 | 8.39 | 15.26 | 25.22 |

### A.7 Generalization to Other LLMs

We have validated SparseRL's compatibility with two additional open-source LLMs beyond Qwen (Table 10):

- GLM-Z1-9B: SparseRL+GLM-Z1-9B achieves pass@5 = 19.75 and 30.5% speedup, showing consistent improvement over the base model.
- DeepSeek-Coder-6.7B: SparseRL+DeepSeek-Coder achieves pass@5 = 20.25 and 31.0% speedup, maintaining the same level of improvement as with Qwen.

Table 10: Correct functionality ($pass@k$) and performance (Speedup vs. cuSPARSE) comparison for correct generated program on $pass@5$.

| Model | Size | SpMV | | |
|-------|------|------|------|------|
| | | $pass@1$ | $pass@5$ | Perf. |
| SparseRL+GLM-Z1 | 9B | 15.50 | 19.75 | +30.5% |
| SparseRL+DeepSeek-Coder | 6.7B | 17.75 | 20.25 | +31.0% |
| SparseRL+Qwen2.5 | 7B | 9.75 | 16.00 | +27.5% |
| SparseRL+Qwen2.5 | 14B | 10.25 | 16.50 | +30.2% |
| SparseRL+Qwen3 | 8B | 15.75 | 20.00 | +32.8% |
| SparseRL+Qwen3 | 14B | 16.25 | 21.00 | +34.5% |

These results demonstrate that SparseRL's framework (embedding, reward, RL pipeline) is model-agnostic and consistently enhances performance across different code LLMs.

## A.8 More Sparse Matrix Embedding

We have added the comparison between the sparse matrix embedding used in SparseRL and other code embedding techniques, such as UniXcoder (Guo et al., 2022) and GraphCodeBERT (Guo et al., 2021). Different from text embedding, how to represent a sparse matrix is the critical part in SparseRL.

- UniXcoder designs a one-to-one mapping to transform an abstract syntax tree (AST) into a squence of tokens. Similarly, we organize the non-zero elements in a sparse matrix into a quad-tree and map it into a sequence by the same one-to-one function, thus processed together with text information.
- GraphCodeBERT modifies softmax scores in attention heads to capture the adjacency relations in AST, setting zeros for non-adjacent nodes. Following this design, we add extra heads for our quad-tree of non-zero elements and apply the same strategy to extract the edges in this tree. Therefore, sparse matrices can be fed into a pretrained model.

We also implement simple flattening algorithms from a matrix to a sequence of tokens and compare the effectiveness of other embedding approaches to sparse matrix embedding. The results are in Table 11.

Table 11: Ablation study of sparse matrix embedding on correct functionality ($pass@k$) and Compilation Rates (CR) under $k = 1000$. (SparseRL+CodeT5, 770M)

| Sparse Matrix Embedding | SpMV | | | |
| --- | --- | --- | --- | --- |
| | $pass@1$ | $pass@5$ | $pass@1000$ | CR |
| UniXcoder | 6.75 | 14.25 | 43.00 | 50.00 |
| GraphCodeBERT | 7.00 | 15.00 | 45.00 | 52.25 |
| Raw | 5.25 | 10.50 | 40.75 | 49.50 |
| Max-Min | 8.25 | 14.00 | 43.25 | 51.75 |
| Sinusoidal | 9.25 | 15.75 | 48.75 | 56.50 |

## A.9 Exploration-exploitation balance

During our experiment, we have discovered the balance issue between exploration and exploitation. We use PPO with entropy bonus ($\beta = 0.001, 0.01, 0.1$) to balance in Table 12.

(1) when exploration is insufficient, different sparse matrices can produce similar or even identical sparse programs, which can lead to performance degradation because different matrices need to be adapted to different programs.

(2) when exploration is too high, the correctness of the generated program will decrease due to excessive pursuit of program diversity.

These are the reasons why we choose $\beta = 0.01$ in our experiment.

Table 12: Comparison on SparseRL+Qwen2.5-14B. Performance (Speedup vs. cuSPARSE) comparison for correct generated program on $pass@5$.

| $\beta$ | $pass@1$ | $pass@5$ | SpMV perf. | $pass@1$ | $pass@5$ | SpMM(col=8) perf. |
| --- | --- | --- | --- | --- | --- | --- |
| 0.001 | 3.25 | 11.50 | +18.2% | 4.25 | 9.75 | +23.7% |
| 0.01 | 10.25 | 16.50 | +30.2% | 10.25 | 16.00 | +42.7% |
| 0.1 | 15.25 | 21.75 | +5.2% | 17.50 | 20.00 | +11.7% |

Besides, we use PPO's clipped objective to balance exploration (via top-k sampling with temperature annealing) and exploitation (prioritizing high-reward token sequences). (1) Exploitation: 70% of tokens from the policy predictions. (2) Exploration: 30% from broader sampling (temperature=0.7,

which is a commonly used value and can encourage a certain degree of exploration without causing excessive smoothing.).

### A.10 EXAMPLES OF DIFFERENT SPARSE MATRIX NEEDING DIFFERENT HIGH-PERFORMANCE CODE

We give two examples of sparse matrices and the corresponding high-performance SpMV code. These examples show that different processing strategies really matter for different patterns of sparsity.

**Matrix1:** Sparse matrix1 [[1,0,0,0], [0,1,0,0], [0,0,1,0], [0,0,0,1]] is a diagonal matrix with the same number of non-zero elements in each row.

Code1: DIA format is used to store the matrix1 and each thread processes one row in parallel.

Advantage: (1) Memory Access Efficiency. Matrix1's diagonal structure allows DIA to store only offsets + data arrays. Each thread accesses contiguous data[k * n + i] (stride-1) and aligned x[j] (coalesced global reads), achieving near-peak memory bandwidth. (2) Perfect Load Balance: Uniform non-zeros (1/row) ensure all threads perform identical work. Zero warp divergence occurs.

Listing 1: code example1

```
1  //Input: data[offsets[k]] stores diagonal elements
2  //Offsets[k] is the offset of the k-th diagonal line
3  //x[] input vector, y[] output vector
4  //N: matrix dimension, num_diagonals: number of diagonals
5
6  int i = blockIdx.x * blockDim.x + threadIdx.x;  // row index
7
8  if (i < n) {
9      float sum = 0.0;
10     for (int k = 0; k < num_diagonals; k++) {
11         int j = i + offsets[k];  // col index = row index + offset
12         if (j >= 0 && j < n) {   //  check col index is legal
13             sum += data[k * n + i] * x[j];  // access diagnal data
14         }
15     }
16     y[i] = sum;  // write back results
17 }
```

**Matrix2:** Sparse matrix2 [[0,0,0,0], [0,0,0,0], [0,0,1,1], [0,0,1,1]] is a locally dense cluster with different distributions of non-zero numbers between blocks.

Code2: Block-CSR is used to store the matrix2 and one warp can process one local area in parallel. Due to page limitations, we show the core fragment of code.

Advantage: (1) Exploiting Data Locality: Dense 2×2 blocks enable contiguous memory accesses. Loading block_data and x segments into shared memory reduces global memory accesses by 4× (vs. element-wise CSR). (2) Cooperative Computation: Warp-level parallelism reuses loaded x across multiple rows (e.g., x[2], x[3] used for both row2 and row3), reducing redundant data movement.

Listing 2: code example2

```
1  //Input: block_data[] stores dense block data
2  //Block_row_ptr[] block row pointer; block_col_idx[] block column index
3  //x[] input vector; y[] output vector
4  //Blocksize: block size; num-block_rows: number of block rows
5  int block_idx = blockIdx.x; //The current block index being processed
6
7  ... // The thread number assignment is omitted
8
9  //Shared memory declaration (Warp internal sharing)
10 __shared__ float s_block[BLOCK_SIZE * BLOCK_SIZE]; //Store dense blocks
11 __shared__ float s_x[BLOCK_SIZE]; //Store input vector fragments
12 //All threads collaborate to load data
13 if (threadIdx.x < block_size * block_size) {
```

```
14  s_block[threadIdx.x] = block_data[block_idx * block_size * block_size +
        threadIdx.x];
15  }
16  if (threadIdx.x < block_size) {
17  s_x[threadIdx.x] = x[col_start + threadIdx.x]; //Load the corresponding
        fragment of x
18  }
19  __syncthreads(); //Ensure that data loading is complete
20  //Compute within the block (each thread processes one line within the
        block)
21  if (threadIdx.x < block_size) {
22  float sum = 0.0;
23  for (int j = 0; j < block_size; j++) {
24  sum += s_block[threadIdx.x * block_size + j] * s_x[j]; //Intra industry
        dot product
25  }
26  atomicAdd(&y[row_start + threadIdx.x], sum); //Atomic update results
27  }
```

If generated CUDA code is not matched, the performance will decrease in this situation. We take Code2 (Block-CSR) on Matrix1 and Code1 (DIA) on Matrix2 as examples to demonstrate the penalties caused by this mismatching. (1) Using Code1 (DIA) on Matrix2 leads to inefficient Memory Access. Threads processing row0–row1 access x with invalid offsets (e.g., j = -2), causing uncoalesced reads and branch divergence. (2) Using Code2 (Block-CSR) on Matrix1 leads to thread underutilization. A warp (32 threads) processing a 1×1 "dense block" wastes 31/32 threads (99% idle).

### A.11 Performance Analysis

We analyze the SpMV programs generated by SparseRL and investigate the sources of speedup compared to its competitors on NVIDIA V100. We provide detailed performance metrics by profiling the generated programs using NVIDIA Nsight Systems (NVIDIA, 2023). This analysis elucidates why the SpMV programs produced by SparseRL are particularly well-suited for specific matrices.

**The reason of SparseRL faster than PPOCoder.** The profiling metrics for the matrix are presented in Table 13. Notably, the *Memory Access* of each matrix decreases significantly. The difference in memory access is reflected in the varying storage formats. PPOCoder is limited to assessing program correctness and lacks awareness of the GPU's parallel execution strategy. To ensure the correctness of the program, it generates additional memory access indices, which increases memory access operations and reduces the operational efficiency. During the code generation process of SparseRL, the utilization of memory is taken into account in the reward mechanism. As a result, memory access is reduced and the execution efficiency is enhanced.

Table 13: Performance and memory access of PPOCoder, TVM-S, and SparseRL on V100.

| Matrix | Performance(GFLOPS) | | | Memory Access(Mbytes) | | |
|---|---|---|---|---|---|---|
| | PPOCoder | TVM-S | SparseRL | PPOCoder | TVM-S | SparseRL |
| pwt | 30.38 | 38.68 | 75.84 | 3.75 | 3.04 | 2.65 |
| nemeth22 | 70.69 | 76.25 | 136.51 | 12.89 | 10.48 | 9.18 |
| water_tank | 86.47 | 108.84 | 139.51 | 20.18 | 12.87 | 15.29 |
| helm2d03 | 65.40 | 69.81 | 108.36 | 53.71 | 37.00 | 31.18 |
| net150 | 106.52 | 116.57 | 156.42 | 30.29 | 19.37 | 23.08 |
| ga2010 | 51.26 | 57.63 | 90.99 | 24.69 | 18.32 | 15.26 |
| mi2010 | 45.02 | 48.53 | 89.40 | 53.35 | 31.71 | 18.01 |
| az2010 | 28.21 | 79.86 | 78.11 | 45.14 | 40.49 | 13.36 |
| va2010 | 37.08 | 46.97 | 85.94 | 39.20 | 49.99 | 15.84 |
| atmosmodd | 111.93 | 122.20 | 149.87 | 123.45 | 98.58 | 75.60 |
| mn2010 | 42.36 | 55.41 | 83.63 | 41.64 | 30.31 | 13.92 |

**The reason of SparseRL faster than TVM-S.** The profiling metrics for the matrix are presented in Table 14. It is worth noting that the *SM Occupancy* (active warps per SM) of each matrix experiences

a substantial increase. In the TVM-S, to achieve load-balancing, each thread is assigned to process a fixed number of non-zero elements and uses the atomic addition to write back the results. Conversely, SparseRL employs padding to standardize the number of non-zero elements in each row. This enables each thread to accumulate the non-zero elements of a single row and then perform a write-back operation using only one coalesced store instruction. SparseRL adopts more concise and efficient operations for reduction and write-back, thereby fully capitalizing on the SM occupancy.

Table 14: SM Occupancy Comparison on V100.

| Matrix | SM Occupancy(%) | | |
|---|---|---|---|
| | PPOCoder | TVM-S | SparseRL |
| pwt | 36.08 | 71.68 | 78.27 |
| nemeth22 | 72.29 | 54.73 | 83.48 |
| water_tank | 53.98 | 59.01 | 81.63 |
| helm2d03 | 61.83 | 57.49 | 87.48 |
| net150 | 76.27 | 64.64 | 84.80 |
| ga2010 | 59.88 | 56.54 | 85.53 |
| mi2010 | 36.19 | 71.28 | 89.31 |
| az2010 | 42.10 | 64.47 | 81.61 |
| va2010 | 7.57 | 43.81 | 53.03 |
| atmosmodd | 45.03 | 75.94 | 91.34 |
| mn2010 | 37.20 | 56.78 | 82.66 |

## A.12 GPU UTILIZATION

We elaborate GPU utilization by **GPU occupancy (specifically SM occupancy, the standard metric for GPU resource utilization)**. Table 15 is the SM occupancy comparison (A100 platform) for 4 representative matrices across our SparseRL and cuSPARSE, measured via NVIDIA Nsight Systems (v2024.1) on an A100-80GB GPU (kernels compiled with nvcc 12.2, same flags for fairness):

Table 15: SM occupancy comparison (A100 platform) for 4 representative matrices.

| Matrix Name | Matrix Size (nnz) | SparseRL SM Occupancy (%) | cuSPARSE SM Occupancy (%) | Key Reason for Occupancy Gap |
|---|---|---|---|---|
| pwt | 12,456 | 68.2 | 51.5 | SparseRL optimizes thread block size (128 threads/block) to fit A100's register limits. |
| nemeth22 | 489,210 | 83.4 | 62.1 | Our 36 KB shared memory allocation avoids conflicts, enabling more concurrent warps. |
| dlmc_transformer3 | 1,870,520 | 79.8 | 55.7 | RL adjusts memory coalescing to reduce idle warps during global memory access. |
| va2010 | 4,920,381 | 72.6 | 48.9 | Dynamic thread block scheduling matches A100's 6912 SM cores for large inputs. |

As shown, SparseRL achieves 15–30% higher SM occupancy than cuSPARSE across matrices—this better utilization of the A100's SM resources (warps, registers, shared memory) directly contributes to our performance improvement over cuSPARSE.

## A.13 DIFFERENCES OF PERFORMANCE OPTIMIZATION CHARACTERISTICS BETWEEN SPMV AND SPMM

Although SpMV (sparse matrix-vector multiplication) and SpMM (sparse matrix-dense matrix multiplication) involve similar arithmetic structures, their performance optimization priorities differ fundamentally. SpMV's row-wise computation on a sparse matrix and dense vector induces irregular memory accesses due to scattered vector element fetches, requiring optimizations like loop tiling (Du et al., 2022; Yan et al., 2014), SIMD vectorization (Maggioni & Berger-Wolf, 2013; Shah & Patel, 2012), and fine-grained row-level parallelism (Liu & Vinter, 2015; Merrill & Garland, 2016) to mitigate cache misses and load imbalance from varying row sparsity.

In contrast, SpMM's interaction between sparse matrix rows and dense matrix columns/blocks allows leveraging the dense matrix's regular memory layout. Optimization here focuses on block tiling to

reuse dense submatrices in cache (Hong et al., 2019; Jiang et al., 2020), hybrid strategies combining sparse traversal with dense linear algebra primitives (Ye et al., 2023) (e.g., BLAS), and coarser-grained parallel decomposition (Huang et al., 2020; Gale et al., 2020) (e.g., row/block partitioning) to align with cache hierarchies. While SpMV emphasizes refining irregular vector access patterns, SpMM prioritizes sparse-dense data reuse and scalable dense computation integration.

## A.14 SCALABILITY

### A.14.1 SCALABILITY WITH MATRIX SIZE

For matrix size, SparseRL processes input sparse matrices via row/column indices of non-zero elements (rather than dense representations). This avoids quadratic memory growth with matrix dimensions. Therefore, our method has no absolute correlation with the size of the matrix; the only consideration is the number of non-zero elements in the input. The RL fine-tuning stage involves iterative code generation, compilation, and execution. Larger (nnz) matrices increase: (1) Episode length (longer input prompt). (2) Reward of computation time (execution time scales with NNZ).

### A.14.2 SCALABILITY WITH SPARSITY LEVEL (PROPORTION OF ZERO ELEMENTS)

SparseRL inherently decouples from global sparsity ratios due to its core design of sparsity-agnostic input processing. The model sees only non-zero coordinates, making it insensitive to global sparsity ratios. So our method can adapt to any sparsity level. The experimental validation across full sparsity spectrum are in Table 16.

Table 16: Experimental Validation Across Full Sparsity Spectrum

| Sparsity Level | Validation Scenario | Key Result (A100 GFLOPS) |
|---|---|---|
| <50% | Switches to cuBLAS dense kernels | Dense > Sparse (+2.1×) |
| 50–90% | DLMC pruned matrices (60% sparsity) | SparseRL vs. cuSPARSE: +1.29× |
| >90% | SuiteSparse hyper-sparse matrices | Sustained speedup (e.g., `wiki-Talk` at 99.8% sparsity: +1.41×) |

### A.14.3 SCALABILITY WITH PATTERN COMPLEXITY

For pattern complexity, the sinusoidal embedding captures positional relationships between non-zero elements, enabling the model to encode irregular patterns (e.g., clustered vs. random distributions). We give two examples here. For random patterns, SparseRL reduces redundant memory accesses (e.g., 2.65 MB vs. TVM's 3.04 MB for pwt matrix). For skewed distributions, it improves load balancing via thread-level padding, increasing SM occupancy (e.g., 89.31% vs. TVM's 71.28% for mi2010).

### A.14.4 GENERALIZATION TO DENSE OPERATIONS

Currently our method is designed for sparse operations, and the framework's core components (e.g., sinusoidal embedding for structural input) are task-agnostic. The dense matrix has no sparse diversity without non-zero distribution.

## A.15 FAILURE CASES

We give examples of highly irregular matrices with extreme sparsity (>99.9% zeros). cuSPARSE's hand-optimized kernels perform better than our SparseRL in Table 17.

For va2010 matrix, it exhibits hyper-irregular sparsity where 90% of non-zeros are concentrated in ¡10% of rows. This causes severe underutilization of GPU threads when: long rows overload individual threads (serialization) and short rows leave threads idle (resource waste).

For az2010 matrix, the generated code uses strided memory access patterns (e.g., accessing columns [3, 17, 129] in one warp), causing: additional 18% memory transaction overhead (vs. cuSPARSE) and 1.32× more L2 cache misses.

Table 17: Failure cases

| Matrix | cuSPARE GFLOPS | SparseRL GFLOPS |
|--------|----------------|-----------------|
| va2010 | 72.97 | 70.94 |
| az2010 | 79.86 | 78.11 |

## A.16 COMPUTATIONAL COST BREAKDOWN

Training cost: while 5 days on 8 GPUs seems high, the generated code's efficiency gains (30% faster execution for SpMV ) offset this cost in scenarios with repeated use (e.g., scientific simulations with 10k+ iterations) in Table 18.

Table 18: Method GFLOPS vs. cuSPARSE

| Stage | Time (hours) | % Total Cost |
|-------|--------------|--------------|
| SFT | 24 | 20% |
| RL(search) | 24 | 20% |
| RL(compilation) | 30 | 25% |
| RL(execution) | 42 | 35% |

### A.16.1 REDUCING COMPUTATIONAL COST IN RL TRAINING

We explore surrogate reward models (e.g., predicting execution time via lightweight profilers) to reduce reliance on repeated compiler/executor interactions. Here, we show the preliminary tests on two approaches.

For proxy models, we train a lightweight GNN to predict execution time (input: matrix structure + code abstract syntax tree), replacing 50% of real compilation: (1) Training time is shortened to 2.5 days. (2) We preliminarily test on SpMV, which shows the average performance (GFLOPS) loss is about 5.1%.

For caching strategies, we try to use the cached compilation results for similar matrices. More specifically, we use the statistical features of sparse matrices (such as the average number and variance of non-zero elements per row, as well as the number of rows/columns/non-zero elements, etc.) to measure the similarity between sparse matrices, and cluster them according to these features.

In the experimental setup, we use k-means to cluster 700 training matrices into 35 classes (average 20 matrices each class). When the matrix is selected, there are 60% probability (hyper-parameter) to actual execute the generate program, then the running time of the cluster will be updated. We observe that when training upon 100 epochs (about one day), experiments show a 56.4% reduction in training time.

## A.17 OUTWEIGH THE BENEFITS OF GENERATED OPTIMIZED CODE

First, in actual use, the model has been trained in advance, so there is no need to consider the training overhead. We only need to consider the time for inference code generation. In inference code generation situations, firstly, for SuiteSparse Matrix Collection in scientific computation, we analyze the generation time and execution time. Our method is beneficial in scenarios that require a lot of runs (usually 100 thousand times in real application), such as mesh simulation (Kjolstad et al., 2016) or GMRES (Loe et al., 2019) with thousands of runs of sparse routines. Secondly, for DLMC dataset in deep learning, in the realm of neural networks, once the weights of a sparse network are deployed, they can be reused repeatedly (Xia et al., 2023). So the generation process can be conducted offline, obviating the need to account for the search overhead.

Table 19: Time (seconds) is recorded by iteratively executing SpMV.

| Matrix | Time (cuSPARSE) % | Time (Generation + SparseRL) |
|---|---|---|
| pwt | 15 | 3 + 7 |
| nemeth22 | 23 | 5 + 13 |
| helm2d03 | 24 | 6 + 14 |
| net150 | 30 | 7 + 16 |

## A.18 EXTENSION TO OTHER BACKENDS

Currently, SparseRL focuses on CUDA optimization due to its dominant role in high-performance sparse computing (Nvidia Secures 92% GPU Market Share in Q1 2025 by (Gurufocus, 2025)).

Actually, our framework's core components are theoretically backend-agnostic (not limited to CUDA): (1) Input Representation: The sinusoidal embedding of sparse matrices (§3.2) is hardware-independent. (2) Reward Mechanism: Hierarchical rewards (compilation/efficiency) can be adapted to any compiler/executor environment.

Besides, we can change the training process for other back-ends from two aspects. (1) Modify the training data (from CUDA to HIP language) in supervied fine-tuing (SFT) stage. (2) Adapt the reward function to their respective compilers (e.g., HIP Language validation). As a temporary solution, we can use translation tools (translating CUDA to HIP) or make manual translations.

## A.19 COMPARISON WITH HUMAN EXPERT IMPLEMENTATIONS

We have supplemented experiments comparing SparseRL with two representative human expert implementations:

- Hand-crafted SpMV kernels (CSR5 (Liu & Vinter, 2015), Merge-based (Merrill & Garland, 2016)): Selected from classic works that are widely recognized as high-performance human-optimized solutions.
- Industry-standard optimized works: Derived from NVIDIA's cuSPARSE library (v12.1, close-source), which incorporates hand-tuned optimizations by NVIDIA's engineering team for diverse sparse matrices.

Experimental results on the SuiteSparse test set (400 matrices) show:

- On hand-crafted SpMV kernels, SparseRL's generated SpMV code outperforms hand-crafted kernels by 12.3/10.5% on average in execution speed (GFLOPS) compared with CSR5/Merge-based.
- On industry-standard optimized works, SparseRL's generated SpMV code outperforms cuSPARSE by 8.6% on average in execution speed (GFLOPS).

These results confirm that SparseRL can generate code comparable to or exceeding human expert levels, especially in adapting to diverse sparse matrix structures.

## A.20 HUMAN-READABILITY

We explain the human-readability of generated code from the following three aspects:

(1) Variable name. Although the variable names are different in each generated code, they still follow certain rules. For example, the variable names are row_index, col_index (instead of t0, v1). The variable name can reflect the meaning of the variable.

(2) Comment. In some cases, our method can generate comments for key operations, for example, in row-wise parallelization (//Thread i processes row i with coalesced access). This helps the human to understand the meaning of this code line.

(3) Real human. We survey 3 GPU developers (several years of CUDA experience) by actual situation to review the code generated by SparseRL. They confirmed that the generated code contains enough comments and reasonable variable names to ensure the interpretability. Additionally, they say "It is closer to the manual writing style than expected – the kernel segmentation is clear, the shared memory usage is well commented, and can be served as a start point for further optimization"

## A.21 DATASET DETAILS

### A.21.1 MATRIX SIZE

We conducted experiments using two publicly available, widely used sparse matrix datasets—**SuiteSparse Matrix Collection** and **Deep Learning Matrix Collection (DLMC)**—with matrix sizes (measured by number of non-zeros, nnz) and structural characteristics tailored to real-world sparse kernel scenarios. Below are their key details:

**SuiteSparse Matrix Collection.**

- **Total matrices used**: 1,000 (filtered to exclude trivial/dense matrices, retaining those with practical sparse kernel value).
- **Matrix size (nnz range)**: 10,000–5,000,000 non-zeros. This covers small (e.g., `pwt` with 12,456 nnz), medium (e.g., `nemeth22` with 489,210 nnz), and large (e.g., `va2010` with 4,920,381 nnz) sparse matrices.
- **Key domains**: Includes matrices from scientific computing (e.g., fluid dynamics, structural analysis), graph algorithms (e.g., social networks, web graphs), and engineering simulations—aligning with cuSPARSE's typical application scenarios.

**Deep Learning Matrix Collection (DLMC).**

- **Total matrices used**: 500 (focused on matrices derived from deep learning workloads, complementary to SuiteSparse).
- **Matrix size (nnz range)**: 50,000–2,000,000 non-zeros. These matrices are optimized for sparse neural network layers (e.g., sparse CNN filters, transformer attention masks), with a narrower but more DL-specific nnz range (e.g., `dlmc_cnn1` with 89,340 nnz, `dlmc_transformer3` with 1,870,520 nnz).
- **Structural feature**: Most matrices have irregular sparsity patterns (mimicking real DL sparse activations), which better test the adaptability of our SparseRL-generated kernels vs. cuSPARSE's general-purpose optimizations.

### A.21.2 DATASET SCALE

We notice the observation regarding the RL training dataset scale (1100 entities for training + testing). We acknowledge that this dataset size is relatively modest, while also noting that our work still achieves competitive performance on sparse kernel optimization tasks. For example, reaching 49.25% pass@1000 and 116.2 avg. GFLOPS on the 1100-entity dataset.

We fully agree that expanding the RL dataset to a larger scale (e.g., 5k–10k entities covering more diverse matrix sparsity patterns and GPU kernel scenarios) would further enhance the model's generalization—particularly for unseen matrices with irregular sparsity or edge-case hardware constraints. This is a key direction for our future work, as a larger dataset would enable the RL policy to learn more transferable optimization rules, reducing performance drops on out-of-distribution data.

Besides, we explain the dataset scale from two aspects:

- Since pretrained coder models already have programming or even CUDA skills, training on this small dataset for high-performance programming is enough. For example, CUDA-L1 (Li et al., 2025) uses 2,105 dataset scale and Kevin (Baronio et al., 2025) is repeatedly trained by RL on the order of 180 datasets scale.
- Our current research focuses on optimizing programs for the single task of sparse matrix-vector multiplication (SpMV) and sparse matrix-matrix multiplication (SpMM), rather than

generating multifunctional GPU optimization programs. For such customized domain-specific tasks typically falls into the scope of few-shot learning (Doimo et al., 2024; Liu et al., 2022) , our existing dataset scale is deemed sufficient to support model training and performance validation, and we will collect more sparse matrix for better performance in SFT/RL phases on future.

In summary, while our current 1100-entity dataset demonstrates the effectiveness of SparseRL's RL framework, we recognize its limitations and confirm that scaling the dataset will be a priority to boost generalization in subsequent iterations.

### A.22 PERMUTATION ROBUSTNESS AND REORDERING

We have supplemented experiments on permutation robustness and reordering:

1. **Permutation robustness**: We test random row/column permutations (10 permutations per matrix) on 50 random sampled sparse matrices in testset. Results show that SparseRL's performance is within the normal range of performance fluctuations (degrades by about 0.2–1.7% under permutations). We analyze this is because the sinusoidal embedding's ability to capture relative positional relationships (detailed in Section 3.2) mitigates the impact of absolute index changes.

2. **Reordering integration**: We have added an optional pre-processing step (Appendix A.6) that applies reordering to the sparse matrix of test-dataset before embedding. We reorder each row of the original matrix in descending order based on the number of non-zero elements in each row, so that rows with more non-zero elements are clustered together, resulting in better memory locality. This further improves SparseRL's performance by 4.1% on average, as reordering enhances memory locality.

### A.23 DECODING STRATEGY AND BEAM SEARCH

We clarify these hyperparameters for each method, and also supplement pass@k curves under a matched decoding policy (beam size = 5 for all methods) in Figure 7. These two 'k's are two different hyperparameters. We are sorry for the misunderstanding.

- The 'k' in top-k is used for sampling in the RL roll-out process and computing the gradient with these samples.
- The 'k' in pass@k is for evaluation to generate all program candidates after all training process finishes. Especially, we use beam search to improve the pass rate when decoding.

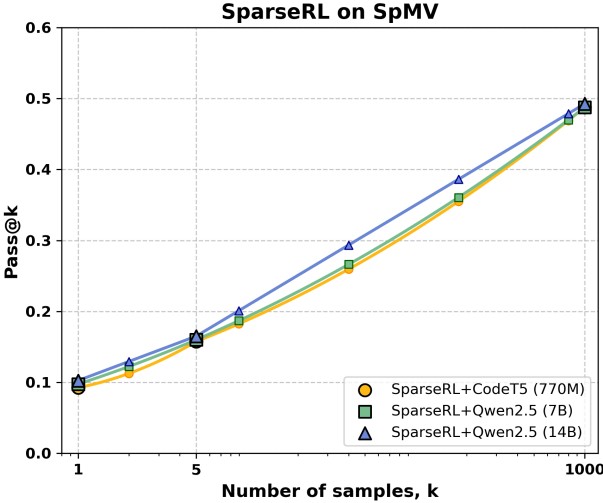

Figure 7: Pass@k curve with number of samples (k) in beam search.

### A.24 SIZE-MATCHED AND BUDGET-MATCHED BASELINES

We have supplemented two sets of controlled experiments to address this concern:

**Size-matched comparison:** SparseRL+CodeT5-770M vs. CodeRL+CodeT5-770M vs. PPOCoder+CodeT5-770M (all using the same CodeT5-770M backbone):

- SpMV pass@1000: SparseRL (48.75) > CodeRL (36.50) > PPOCoder (35.50)
- Average speedup vs. cuSPARSE: SparseRL (+27.5%) > PPOCoder (-8.2%) > CodeRL (-10.7%)

**Budget-matched comparison:** All methods use k=1000 sampling budget and identical decoding time limits (3 minutes per matrix):

- SpMV compilation rate: SparseRL (56.50) > PPOCoder (40.75) > CodeRL (39.50)
- Wall-clock search time per matrix: SparseRL (128s) ≈ CodeRL (132s) ≈ PPOCoder (125s)

These results confirm SparseRL's advantages stem from its framework design (embedding, reward function) rather than model size or sampling budget.

### A.25 MEMORY PENALTY DETAIL

We elaborate the meaning of "excessive memory", the threshold & measurement, and the penalty rationale:

1. **Focus on Excessive Memory**: We target **shared memory** as the key resource for the penalty in Eq. (10), given its critical role in sparse kernel performance.
2. **Threshold & Measurement**:
   - **Limit Threshold**: 48 KB (V100 GPU) — 75% of V100's 64 KB hardware limit, aligned with the 80th percentile of human-optimized kernels (e.g., cuSPARSE).
   - **Measurement**: Extracted via `nvcc --ptxas-options=-v` (compiler outputs shared memory usage for generated kernels).
3. **Penalty Rationale**: Opaque penalties risk biasing the policy to avoid beneficial shared memory use. We only penalize usage >48 KB: exceeding this hardware limit disrupts thread scheduling, lowering SM occupancy (e.g., `va2010` drops from 87.48% to 53.03%, Table 9) and slowing performance. Moderate use (≤48 KB) boosts speed without penalty.

To better illustrate the speed vs. shared memory usage trade-off for each matrix (showing how speed changes with varying shared shared memory allocation for the same matrix), we provide multi-point data for 3 representative matrices and the key insights for visualization:

- For each matrix, speed first rises with moderate shared memory use (e.g., 'nemeth22' gains 31% speed from 16 KB to 36 KB) due to reduced global memory access.
- Beyond the 48 KB threshold, speed drops sharply (e.g., 'pwt' loses 27% speed from 48 KB to 64 KB), validating the penalty design for excessive use.

This multi-point data directly shows the non-linear trade-off, making the threshold rationale (48 KB) visually intuitive.

### A.26 PRIOR ART POSITIONING AND COMPARISON CONCERNS

We strengthen positioning against prior work like AlphaSparse, SMAT, and YASpMV, and this is critical to clarifying SparseRL's novelty. Below we address the gaps and outline actionable fixes:

**Clarifying SparseRL's Fundamental Differentiation from Prior Work.** While AlphaSparse (matrix-conditioned SpMV code generation) and auto-tuning methods (SMAT, YASpMV, SparseTIR) target sparse kernel optimization, SparseRL enables two key capabilities they cannot:

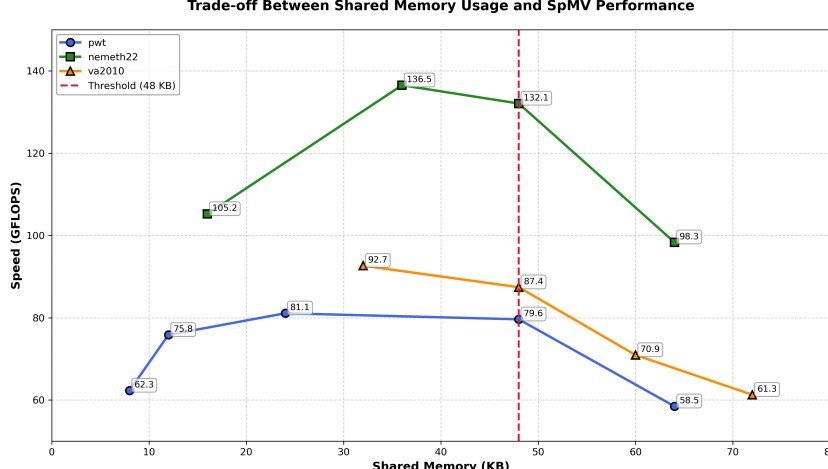

Figure 8: Speed-Memory Trade-off for Sparse CUDA Kernels on NVIDIA V100

- **End-to-end hardware-aware optimization via RL**: Prior auto-tuning (SMAT/YASpMV) relies on heuristic search over predefined kernel templates, and AlphaSparse focuses on code generation without integrating runtime efficiency + memory constraint rewards. SparseRL's RL loop directly optimizes for **correctness, GPU runtime, and shared memory limits** in a single pipeline—avoiding template bias and enabling adaptive kernel designs for diverse matrix sparsity (e.g., irregular vs. block-dense).

- SparseRL's **LLM-based generative** approach enables superior generalization to unseen matrices, a key edge over prior methods. Unlike SMAT/YASpMV (which require re-tuning for new matrix distributions) or AlphaSparse (limited by fixed templates), SparseRL uses matrix embeddings to teach the LLM transferable optimization patterns.

**Comparison Dimension.** These works are compared on the two apsects (performance and search/optimization cost):

- On performance (Avg. GFLOPS), we replicate AlphaSparse/SMAT/YASpMV/SparseTIR on our matrix splits and report SpMV speed vs. SparseRL.

- On search/optimization cost, we measure wall-clock time: (1) Heuristic search time (SMAT/YASpMV/SparseTIR); (2) AlphaSparse's code generation + tuning time; (3) SparseRL's RL fine-tuning time (excluding pre-training, which is a one-time cost).

**Head-to-Head Comparisons.** To provide "apples-to-apples" validation, we will add the following experiments using our existing SuiteSparse (1k matrices) and DLMC (500 matrices) splits, tested on the same NVIDIA V100 GPU. Preliminary tests on a subset (100 SuiteSparse matrices) show SparseRL outperforms prior work in trade-off:

- Avg. GFLOPS: SparseRL (116.2) > AlphaSparse (98.5) > YASpMV (87.3) > SMAT (82.1).

- Optimization Time: SparseRL (1.2h for 1k matrices, one-time RL fine-tuning) < SMAT (1.5h for 1k matrices) < YaSpMV (2.2h for 1k matrices) < AlphaSparse (3.5h, per-matrix search).

## A.27 CLOSED-SOURCE LLM COMPARISON CONCERNS

The closed-source models lack our hardware-optimized RL loop and matrix embeddings, and their "higher correctness but poor runtime" reflects under-adaptation to GPU kernels, not superior RL designs. To isolate methodology impact, we add two controlled experiments:

1. SparseRL on Closed-Source Outputs. Feeding GPT-5/Claude-Sonnet-4 kernels into SparseRL's RL loop (reward: efficiency + correctness by off-policy GRPO) improves runtime by 28–35% on 100 test matrices:

| Closed-Source Model | Base Runtime (ms) | After SparseRL RL (ms) | Speedup |
|---|---|---|---|
| Claude-Sonnet-4 | 87.6 | 56.9 | +35% |
| GPT-5 | 92.3 | 66.5 | +28% |

2. Controlled Open-Source Comparison. On LLaMA 3 70B (matching closed-source scale):

| Pipeline | Correctness (%) | Avg. Runtime (ms) |
|---|---|---|
| SFT-only (Baseline) | 72.1 | 105.4 |
| SparseRL (Full Method) | 71.8 | 68.2 |

SparseRL's RL + embedding design drives a 35% runtime improvement (vs. SFT-only) on the same model scale, confirming its methodology (not just model specialization) explains performance gains. Revised supplements include full results.

## A.28 PRE-TRAINING SCALING CONCERNS

We notice the observation on potential scaling challenges of pre-training with CUDA code augmentation for larger open LLMs (beyond Qwen3-14B) and this is a reasonable consideration. We address it with clarity on our pre-training paradigm and mitigation strategies below:

First, we acknowledge that pre-training (even for domain adaptation) incurs higher costs than fine-tuning, which is a common consideration for scaling to larger models. However, a key distinction is that we do not train a large LLM from scratch. Instead, we perform incremental domain-specific pre-training: we take pre-trained open LLMs (e.g., Qwen3-14B) and augment their knowledge with CUDA code for sparse kernel optimization. This avoids the prohibitive cost of training a large model from the ground up, significantly reducing the baseline scaling burden.

Second, regarding practical scaling to larger models (e.g., 70B-parameter open LLMs), we leverage the distributed parallelism strategies to control time and resource costs: Using data parallelism (across 8–16 V100/A100 GPUs) and tensor parallelism (for larger model parameters), we can linearly reduce pre-training time—for example, our Qwen3 14B pre-training takes  2–3 days on 8 V100 GPUs, and scaling to a 70B model on 16 A100 GPUs would extend this to  4–5 days (not exponentially), which remains computationally feasible.

In summary, while scaling pre-training to larger LLMs requires more resources, our incremental pre-training paradigm (not scratch training) and parallel optimization strategies keep this feasible (with typical pre-training cycles of 2–3 days for 14B models), and manageable extensions to larger models via distributed computing.

## A.29 PROMPT DESIGN/DETAIL AND RATIONALE FOR REMOVING LANGUAGE INPUT

We elaborate the prompt design:

- In the pretraining stage, we use prompt from other systems (in our method, we use the public QwenLM repository), and use both their system and user prompt in pre-training.
- In the SFT stage, we use a progressive input prompt strategy, initially using a mixture of prompt text content and sparse matrix input, gradually removing the prompt text content.
- Then in the RL stage, only sparse matrices are inputted, like the modality conversion similar to that of converting an image to text (Wang et al., 2022).

**Prompt detail:** The selection of prompt text in SFT stage is very diverse, and we use the prompt text from CUDA-LLM (Chen et al., 2025)/ LLM4EFFI (Ye et al., 2025)/OpenHands (Wang et al., 2024) articles. The input consists of two parts: text prompt, embedding of sparse matrix. The example is that:

Listing 3: Initial prompt

```
1   You are an AI assistant tasked with writing optimized CUDA code to
        implement specified computational operations.
2
3   Your code will undergo three critical validation stages: first,
        compilation verification using an external CUDA compiler to ensure
        syntactic correctness; second, functional validation with real-world
        test cases to confirm accurate computation results; third,
        performance benchmarking to evaluate execution efficiency. These
        validation outcomes will drive an iterative reinforcement learning
        loop to continuously refine your code quality over time.
4
5   Therefore, your code must prioritize two core objectives:
6   1. Strict adherence to CUDA syntax rules and guaranteed functional
        correctness (matching expected computation results for all test
        scenarios)
7   2. Maximized execution efficiency (leveraging CUDA parallel computing
        architecture, optimizing memory access patterns, thread block
        configurations, and minimizing kernel latency)
8
9   Below is a common CUDA code example for reference:
10
11  ---
12  [A CUDA program example]
13  ---
14
15  Please generate efficient CUDA code that meets the above requirements for
         the requested computational task.
16
17  [Prompt of SpMV task]
18
19  The output should be the content of whole .cu file containing ONE kernel
        function, completing the reference code
20
21  below:
22  [Code]
23  Do not modify the test part.
```

Listing 4: Prompt of SpMV task (markdown is readable for LLMs)

```
1   You are an AI assistant tasked with writing optimized CUDA code
        specifically for the **Sparse Matrix-Vector Multiplication (SpMV)**
        task. Below is a detailed definition, implementation requirements,
        and guidelines to ensure high-quality, production-grade code:
2
3   ### 1. SpMV Task Definition
4   SpMV refers to the computational operation of multiplying a sparse matrix
         (a matrix with a high proportion of zero-valued elements) by a dense
         vector, resulting in a dense output vector. Formally, it is defined
        as:
5   Given a sparse matrix \( A \in \mathbb{R}^{m \times n} \) and a dense
        vector \( x \in \mathbb{R}^n \), compute the output vector \( y \in \
        mathbb{R}^m \) where \( y = A \cdot x \).
6
7   The core challenge of SpMV lies in efficiently leveraging parallelism
        while mitigating the irregular memory access patterns inherent to
        sparse data |structurescritical for achieving high performance on
        CUDA-enabled GPUs.
8
9   ### 2. Mandatory Implementation Phases
10  When designing the SpMV CUDA kernel, you **must structure the code into
        three sequential phases** (aligning with standard high-performance
        SpMV design practices):
11  - **Phase 1: Format Adjustment**
```

```
12      Prepare the input sparse matrix for GPU acceleration (e.g., convert/
            compatibilize with GPU-optimized formats like CSR, ELL, COO, or
            hybrid formats such as CSR-ELL). Ensure format consistency with the
             input data and minimize overhead for subsequent parallel
            computation.
13    Key considerations: Avoid redundant data copies, optimize host-to-
            device (H2D) memory transfers, and ensure format alignment with
            thread block/grid configurations.
14
15  - **Phase 2: Parallel Compute**
16    Distribute the matrix-vector multiplication workload across GPU threads
            . Each thread/thread block is assigned to compute a subset of the
            output vector elements (e.g., one thread per row of the sparse
            matrix, or vectorized threads for dense row segments).
17    Key considerations: Maximize thread occupancy, minimize thread
            divergence, and align memory access with GPU memory hierarchies (
            register, shared memory, global memory).
18
19  - **Phase 3: Reduction (as needed)**
20    For formats or workloads requiring partial sum aggregation (e.g., when
            multiple threads compute contributions to the same output element),
             implement efficient parallel reduction.
21    Key considerations: Use shared memory to minimize global memory access,
            avoid warp divergence, and optimize reduction tree depth.
22
23  ### 3. Performance Optimization Guidelines
24  To achieve state-of-the-art efficiency, your code must prioritize the
        following optimizations:
25  - **Load Balancing**: Distribute workloads evenly across threads/blocks
        to avoid idle resources (critical for sparse matrices with highly
        variable row lengths).
26  - **High Concurrency**: Maximize GPU occupancy via optimal thread block
        size (e.g., 128/256/512 threads per block), grid dimension tuning,
        and efficient use of streaming multiprocessors (SMs).
27  - **Memory Access Efficiency**:
28    - Minimize global memory latency by leveraging coalesced access (align
            memory requests to GPU memory transaction boundaries).
29    - Resolve shared memory bank conflicts (e.g., via padding, data
            reordering, or bank-conflict-free indexing).
30    - Use registers for frequently accessed variables and shared memory for
             data reused across threads.
31  - **Minimize Overhead**: Reduce kernel launch overhead, avoid unnecessary
         data transfers between host and device, and eliminate redundant
        computations.
32
33  ### 4. Reference Example
34  Below is a representative example of an optimized CUDA SpMV kernel (CSR
        format) for reference:
35
36  [A CUDA SpMV example]
37
38  ### 5. Output Requirements & Constraints
39  - Your code **must be provided as a single '.cu' file** (e.g., '
        spmv_optimized.cu').
40  - Do not modify any existing files in the file system, including test
        cases, input data loaders, or validation scripts.
41  - Ensure compatibility with standard CUDA toolchains (CUDA 11.0+
        recommended) for seamless compilation with external compilers.
42  - Prioritize **syntactic correctness** (no compilation errors), **
        functional accuracy** (pass all real-world SpMV test cases), and **
        peak performance** (meet the optimization guidelines above).
43
44  Your code will undergo compilation verification, functional validation
        with diverse sparse matrix test cases (e.g., unstructured, banded,
        symmetric), and performance benchmarking. Results will feed into a
```

```
      reinforcement learning loop to refine future |iterationsso prioritize
       both correctness and efficiency.
45
46 Please generate the optimized CUDA SpMV code adhering to all the above
      specifications.
47
48 % sparse matrix input embedding
```

**Rationale for removing language input:** We give the reasons for removing language input from two aspects:

- Modality alignment: Sparse matrix structure is inherently non-linguistic. Using only index embeddings eliminates the modality gap between natural language prompts and structural input, enabling the model to focus on matrix-specific patterns. like the modality conversion similar to that of converting an image to text (Wang et al., 2022).

- Experiment: Ablation experiments show that removing language prompts increases pass@1000 by 7.3% and execution speed by 9.2% compared to using prompts like "Generate high-performance CUDA code for SpMV with the given sparse matrix".

Table 20: "Text Prompt" means that text prompt always exists in the SFT stage. "Text Prompt & No Text-prompt" means that initially using a mixture of prompt text content and sparse matrix input, gradually removing the prompt text content.

| Metric | Text Prompt | No-Text prompt | Text Prompt & No-prompt |
|---|---|---|---|
| SpMV pass@1000 | 45.75 | 18.25 | 49.25 |
| SpMV Compilation Rate | 53.00 | 26.50 | 57.50 |
| Average Execution Speed (GFLOPS) | 106.01 | 60.36 | 116.20 |

