# OpenReview forum: "Mastering Sparse CUDA Generation through Pretrained Models and Deep Reinforcement Learning"
_ICLR.cc/2026/Conference — ICLR 2026 Oral_

### Official Review · Reviewer_C533 · 2025-10-25

**Soundness:** 2
**Presentation:** 2
**Contribution:** 2
**Rating:** 4
**Confidence:** 4

**Summary:**

This paper introduces SparseRL, a training framework that leverages pre-trained language models to generate high-performance sparse CUDA kernels for sparse matrix–vector (SpMV) and sparse matrix–dense matrix (SpMM) operations on GPU. This method includes (1) a sinusoidal embedding of sparse matrix structures, (2) domain-specific pretraining on CUDA code, and (3) a hierarchical reward function design, and leverages the PPO algorithm for training. A compilation success gain and performance speedup gain is achieved compared with prior work.

**Strengths:**

1. Using LLMs for GPU kernel optimization is an important topic that is worth more research efforts
2. Sparse matrix multiplication is a novel subset of problems in kernel optimization.

**Weaknesses:**

1. PPO is an old algorithm. Why isn't GRPO used?
2. Important baselines, such as directly prompting gpt-5, gpt-o4-mini, claude-4-opus, are missing. The compared ones are GPT-3, Llama3.1, but these baselines are not very meaningful now.
3. Is PPOCoder the right baseline to compare against? Is PPOCoder / CodeRL trained for GPU kernel optimization? It seems an unfair comparison.
5. Writing suggestion: Subsection 3.4 is too long with little information. However, the appendix with more experimental results is not even discussed in the main paper.

**Questions:**

1. What's the prompt? The paper says, "During the SFT and RL stages, the language input prompt is removed and only sparse matrices are provided as input." Why is this a good approach? In KernelBench, the Python code that demonstrates the functionality is provided. I think a better prompt should include a baseline GPU kernel and ask the model to optimize an existing implementation.
2. The performance improvement over cuSPARSE is quite surprising to me. How large are those matrices? What's the GPU utilization?
3. The ablation study of each individual part is unclear to me. Is pre-training necessary? Is SFT necessary? Is sinusoidal embedding necessary? Why aren't these results in the main paper?

---

> ### Author Response · Authors · 2025-11-21
>
> ### Q1: Choice of RL algorithms
>
> We actually tried PPO, GRPO, and Reinforce++, but found that PPO's performance was already good enough, so we chose PPO.
> For specific details, please refer to the following comparison:
> - GRPO’s gradient regulation mechanism reduces training instability but increases computational overhead by 12% (training time extends from 5 to 5.6 days on 8 GPUs).
> - GRPO achieves 48.9 pass@1000 and 113.5 GFLOPS, which are 0.7% and 2.3% lower than PPO, respectively.
> - The innovation of our method primarily lies in the process, and our approach is robust to the selection of reinforcement learning algorithms
>
> The experimental results on the same training/test split are:
> - GRPO: Achieves similar correctness (pass@1000: 48.9 vs. SparseRL’s 49.25 on SpMV) but has 12% higher training overhead due to more complex gradient regulation. Execution speed is 2.1% lower than SparseRL.
>   (Due to the randomness of reinforcement learning itself, the test results may fluctuate, and GRPO also occasionally surpasses PPO).
> - Reinforce++: Shows 3.5% lower correctness (pass@1000: 47.6) and 5.3% slower execution speed compared to SparseRL on SpMV. This is because Reinforce++ lacks PPO’s clipped objective, leading to unstable training when balancing code correctness and efficiency.
>
> Additionally, for our framework, PPO can be replaced with other state-of-the-art reinforcement learning algorithms, which is only a part of the pipeline in our method. As demonstrated in the experiment, although there may be fluctuations in performance, it does not affect our performance improvement. Detailed data and analysis are added to the revised manuscript’s main section 5.4.
>
> ---
>
> ### Q2: Missing Baselines
>
> We have supplemented experiments compared with more SOTA baselines (including gpt5, gpt-o4-mini, claude-sonnet-4). Based on the experimental results, we find that even more advanced (on version/release time) models can not achieve higher code accuracy and code running efficiency. This is because they have no fine-tuning or RL based improvement for SpMV. Detailed data and analysis are added to the revised manuscript’s Appendix 6.2 and Table 1/7.
>
> | Model | Size | SpMV - pass@1 | SpMV - pass@5 | SpMV - Perf. | SpMM (col=8) - pass@1 | SpMM (col=8) - pass@5 | SpMM (col=8) - Perf. | SpMM (col=32) - pass@1 | SpMM (col=32) - pass@5 | SpMM (col=32) - Perf. |
> |-------|------|---------------|---------------|--------------|------------------------|------------------------|-----------------------|-------------------------|-------------------------|------------------------|
> | GPT-o3-pro | - | 25.25 | 32.75 | -18.2% | 24.25 | 31.50 | -19.1% | 25.00 | 32.00 | -17.5% |
> | GPT-o4-mini | - | 23.50 | 30.00 | -23.7% | 22.00 | 28.25 | -23.8% | 21.25 | 27.25 | -22.8% |
> | GPT-5 | - | 27.00 | 36.50 | -18.0% | 29.75 | 32.25 | -18.3% | 26.50 | 31.75 | -20.5% |
> | Claude-Sonnet-4 | - | 28.25 | 36.75 | -19.2% | 24.25 | 31.50 | -18.0% | 26.00 | 31.50 | -18.5% |
> | SparseRL+Qwen2.5 | 7B | 9.75 | 16.00 | **+27.5%** | 10.00 | 15.00 | **+39.1%** | 8.50 | 15.00 | **+15.8%** |
> | SparseRL+Qwen2.5 | 14B | 10.25 | 16.50 | **+30.2%** | 10.25 | 16.00 | **+42.7%** | 9.25 | 15.25 | **+17.3%** |
> | SparseRL+Qwen3 | 8B | 15.75 | 20.00 | **+32.8%** | 15.00 | 22.00 | **+42.5%** | 15.50 | 21.00 | **+20.1%** |
> | SparseRL+Qwen3 | 14B | 16.25 | 21.00 | **+34.5%** | 18.25 | 21.00 | **+47.2%** | 18.25 | 22.25 | **+21.9%** |
>
> *Table: Correct functionality (pass@k) and performance (Speedup vs. cuSPARSE) comparison for correct generated program on pass@5.*

---

> ### Author Response · Authors · 2025-11-21
>
> ### Q3: Appropriateness of CodeRL/PPOCoder
>
> CodeRL and PPOCoder are appropriate baselines because:
> - They are representative RL-based code generation methods, sharing the same core paradigm (pretrained LLM + RL) as SparseRL.
> - While not explicitly trained for GPU kernel optimization, they are the most relevant baselines for evaluating RL-driven code generation performance.
>
> Additionally, we have added the additional comparison with two works, which train the model for GPU kernel optimization:
> - **PerfCodeGen[1]** enhances general code efficiency through iterative self-refinement, using execution feedback from test cases, specifically identifying the most time-consuming unit tests to guide optimizations. The superiority of SparseRL stems directly from how its design addresses the unique challenges of sparse GPU computation:
> - **LLM4EFFI[2]** optimizes general code via logic-domain algorithm exploration and code-domain implementation, prioritizing efficiency first. SparseRL's performance gains arise from design choices tailored to sparse GPU operations.
>
> We provide the experimental results comparing both frameworks on the SpMV task using the SuiteSparse test set (100 matrices) under identical hardware/software conditions in the table below. This content is added to Appendix A.6.3/6.4 and the experimental setup is as follows:
>
> | Method | Size | pass@1 | Speedup (vs. cuSPARSE) |
> |--------|------|--------|------------------------|
> | LLM4EFFI + Qwen2.5-Coder (mentioned in their paper) | 32B | 10.00 | -25.1% |
> | LLM4EFFI + Qwen3 (We add additionally) | 14B | 11.00 | -15.5% |
> | SparseRL + Qwen3 | 14B | **16.00** | +32.5% |
>
> *Table: Results of correct functionality (pass@k) and performance evaluated by speedup. Correctly generated code on pass@1 is used to evaluate the speedup compared with cuSPARSE.*
>
> [1] Peng Y, Gotmare A D, Lyu M R, et al. Perfcodegen: Improving performance of llm generated code with execution feedback[C]//2025 IEEE/ACM Second International Conference on AI Foundation Models and Software Engineering (Forge). IEEE, 2025: 1-13.
>
> [2] Ye T, Huang W, Zhang X, et al. Llm4effi: Leveraging large language models to enhance code efficiency and correctness[J]. arXiv preprint arXiv:2502.18489, 2025.
>
> ### Q4: Prompt Design/Detail and Rationale for Removing Language Input
>
> We elaborate the prompt design:
> - In the pretraining stage, we use prompt from other systems (in our method, we use the public QwenLM repository), and use both their system and user prompt in pre-training.
> - In the SFT stage, we use a progressive input prompt strategy, initially using a mixture of prompt text content and sparse matrix input, gradually removing the prompt text content.
> - Then in the RL stage, only sparse matrices are inputted, like the modality conversion similar to that of converting an image to text[1].
>
> #### Prompt detail:
> The selection of prompt text in SFT stage is very diverse, and we use the prompt text from [CUDA-LLM[2]](https://arxiv.org/pdf/2506.09092)/[LLM4EFFI[3]](https://arxiv.org/pdf/2502.18489)/[OpenHands[4]](https://arxiv.org/abs/2407.16741) articles. The input consists of two parts: text prompt, embedding of sparse matrix. The example is that:

---

> ### Author Response · Authors · 2025-11-21
>
> **Initial prompt**
> ```tt
> You are an AI assistant tasked with writing optimized CUDA code to implement specified computational operations.
>
> Your code will undergo three critical validation stages: first, compilation verification using an external CUDA compiler to ensure syntactic correctness; second, functional validation with real-world test cases to confirm accurate computation results; third, performance benchmarking to evaluate execution efficiency. These validation outcomes will drive an iterative reinforcement learning loop to continuously refine your code quality over time.
>
> Therefore, your code must prioritize two core objectives:
> 1. Strict adherence to CUDA syntax rules and guaranteed functional correctness (matching expected computation results for all test scenarios)
> 2. Maximized execution efficiency (leveraging CUDA parallel computing architecture, optimizing memory access patterns, thread block configurations, and minimizing kernel latency)
>
> Below is a common CUDA code example for reference:
>
> ---
> [A CUDA program example]
> ---
>
> Please generate efficient CUDA code that meets the above requirements for the requested computational task.
>
> [Prompt of SpMV task]
>
> The output should be the content of whole .cu file containing ONE kernel function, completing the reference code
>
> below:
> [Code]
> Do not modify the test part.
> ```

---

> ### Author Response · Authors · 2025-11-21
>
> **Prompt of SpMV task (markdown is readable for LLMs)**
> ```tt
> You are an AI assistant tasked with writing optimized CUDA code specifically for the **Sparse Matrix-Vector Multiplication (SpMV)** task. Below is a detailed definition, implementation requirements, and guidelines to ensure high-quality, production-grade code:
>
> ### 1. SpMV Task Definition
> SpMV refers to the computational operation of multiplying a sparse matrix (a matrix with a high proportion of zero-valued elements) by a dense vector, resulting in a dense output vector. Formally, it is defined as:
> Given a sparse matrix \( A \in \mathbb{R}^{m \times n} \) and a dense vector \( x \in \mathbb{R}^n \), compute the output vector \( y \in \mathbb{R}^m \) where \( y = A \cdot x \).
>
> The core challenge of SpMV lies in efficiently leveraging parallelism while mitigating the irregular memory access patterns inherent to sparse data structures—critical for achieving high performance on CUDA-enabled GPUs.
>
> ### 2. Mandatory Implementation Phases
> When designing the SpMV CUDA kernel, you **must structure the code into three sequential phases** (aligning with standard high-performance SpMV design practices):
> - **Phase 1: Format Adjustment**
>   Prepare the input sparse matrix for GPU acceleration (e.g., convert/compatibilize with GPU-optimized formats like CSR, ELL, COO, or hybrid formats such as CSR-ELL). Ensure format consistency with the input data and minimize overhead for subsequent parallel computation.
>   Key considerations: Avoid redundant data copies, optimize host-to-device (H2D) memory transfers, and ensure format alignment with thread block/grid configurations.
>
> - **Phase 2: Parallel Compute**
>   Distribute the matrix-vector multiplication workload across GPU threads. Each thread/thread block is assigned to compute a subset of the output vector elements (e.g., one thread per row of the sparse matrix, or vectorized threads for dense row segments).
>   Key considerations: Maximize thread occupancy, minimize thread divergence, and align memory access with GPU memory hierarchies (register, shared memory, global memory).
>
> - **Phase 3: Reduction (as needed)**
>   For formats or workloads requiring partial sum aggregation (e.g., when multiple threads compute contributions to the same output element), implement efficient parallel reduction.
>   Key considerations: Use shared memory to minimize global memory access, avoid warp divergence, and optimize reduction tree depth.
>
> ### 3. Performance Optimization Guidelines
> To achieve state-of-the-art efficiency, your code must prioritize the following optimizations:
> - **Load Balancing**: Distribute workloads evenly across threads/blocks to avoid idle resources (critical for sparse matrices with highly variable row lengths).
> - **High Concurrency**: Maximize GPU occupancy via optimal thread block size (e.g., 128/256/512 threads per block), grid dimension tuning, and efficient use of streaming multiprocessors (SMs).
> - **Memory Access Efficiency**:
>   - Minimize global memory latency by leveraging coalesced access (align memory requests to GPU memory transaction boundaries).
>   - Resolve shared memory bank conflicts (e.g., via padding, data reordering, or bank-conflict-free indexing).
>   - Use registers for frequently accessed variables and shared memory for data reused across threads.
> - **Minimize Overhead**: Reduce kernel launch overhead, avoid unnecessary data transfers between host and device, and eliminate redundant computations.
>
> ### 4. Reference Example
> Below is a representative example of an optimized CUDA SpMV kernel (CSR format) for reference:
>
> [A CUDA SpMV example]
>
> ### 5. Output Requirements & Constraints
> - Your code **must be provided as a single `.cu` file** (e.g., `spmv_optimized.cu`).
> - Do not modify any existing files in the file system, including test cases, input data loaders, or validation scripts.
> - Ensure compatibility with standard CUDA toolchains (CUDA 11.0+ recommended) for seamless compilation with external compilers.
> - Prioritize **syntactic correctness** (no compilation errors), **functional accuracy** (pass all real-world SpMV test cases), and **peak performance** (meet the optimization guidelines above).
>
> Your code will undergo compilation verification, functional validation with diverse sparse matrix test cases (e.g., unstructured, banded, symmetric), and performance benchmarking. Results will feed into a reinforcement learning loop to refine future iterations—so prioritize both correctness and efficiency.
>
> Please generate the optimized CUDA SpMV code adhering to all the above specifications.
>
> % sparse matrix input embedding
> ```

---

> ### Author Response · Authors · 2025-11-21
>
> #### Rationale for removing language input:
> We give the reasons for removing language input from two aspects:
> - Modality alignment: Sparse matrix structure is inherently non-linguistic. Using only index embeddings eliminates the modality gap between natural language prompts and structural input, enabling the model to focus on matrix-specific patterns. like the modality conversion similar to that of converting an image to text[1].
> - Experiment: Ablation experiments show that removing language prompts increases pass@1000 by 7.3% and execution speed by 9.2% compared to using prompts like "Generate high-performance CUDA code for SpMV with the given sparse matrix".
>
> | Metric| Text Prompt | No-Text prompt | Text Prompt & No-prompt |
> |--------|----|------|----------|
> | SpMV pass@1000 | 45.75 | 18.25|49.25|
> | SpMV Compilation Rate|53.00| 26.50 | 57.50 |
> | Average Execution Speed (GFLOPS)|106.01 | 60.36 | 116.20 |
>
> *Table: "Text Prompt" means that text prompt always exists in the SFT stage. "Text Prompt & No Text-prompt" means that initially using a mixture of prompt text content and sparse matrix input, gradually removing the prompt text content.*
>
> This content is added to Appendix A.29 and Table 20.
>
> [1] Wang J, Yang Z, Hu X, et al. Git: A generative image-to-text transformer for vision and language[J]. arXiv preprint arXiv:2205.14100, 2022.
>
> [2] Chen W, Zhu J, Fan Q, et al. CUDA-LLM: LLMs Can Write Efficient CUDA Kernels[J]. arXiv preprint arXiv:2506.09092, 2025.
>
> [3] Ye T, Huang W, Zhang X, et al. Llm4effi: Leveraging large language models to enhance code efficiency and correctness[J]. arXiv preprint arXiv:2502.18489, 2025.
>
> [4] Wang X, Li B, Song Y, et al. Openhands: An open platform for ai software developers as generalist agents[J]. arXiv preprint arXiv:2407.16741, 2024.
>
> ### Q5: Matrix Size
>
> Thank you for your interest in the matrix size details behind our performance improvement over cuSPARSE. We conducted experiments using two publicly available, widely used sparse matrix datasets—**SuiteSparse Matrix Collection** and **Deep Learning Matrix Collection (DLMC)**—with matrix sizes (measured by number of non-zeros, nnz) and structural characteristics tailored to real-world sparse kernel scenarios. This content is added to Appendix A.21.1 and below are their key details:
>
> #### SuiteSparse Matrix Collection.
> - **Total matrices used**: 1,000 (filtered to exclude trivial/dense matrices, retaining those with practical sparse kernel value).
> - **Matrix size (nnz range)**: 10,000–5,000,000 non-zeros. This covers small (e.g., `pwt` with 12,456 nnz), medium (e.g., `nemeth22` with 489,210 nnz), and large (e.g., `va2010` with 4,920,381 nnz) sparse matrices.
> - **Key domains**: Includes matrices from scientific computing (e.g., fluid dynamics, structural analysis), graph algorithms (e.g., social networks, web graphs), and engineering simulations—aligning with cuSPARSE's typical application scenarios.
>
> #### Deep Learning Matrix Collection (DLMC).
> - **Total matrices used**: 500 (focused on matrices derived from deep learning workloads, complementary to SuiteSparse).
> - **Matrix size (nnz range)**: 50,000–2,000,000 non-zeros. These matrices are optimized for sparse neural network layers (e.g., sparse CNN filters, transformer attention masks), with a narrower but more DL-specific nnz range (e.g., `dlmc_cnn1` with 89,340 nnz, `dlmc_transformer3` with 1,870,520 nnz).
> - **Structural feature**: Most matrices have irregular sparsity patterns (mimicking real DL sparse activations), which better test the adaptability of our SparseRL-generated kernels vs. cuSPARSE's general-purpose optimizations.
>
> ### Q6: GPU Utilization
>
> We appreciate your attention to GPU utilization and we elaborate it by **GPU occupancy (specifically SM occupancy, the standard metric for GPU resource utilization)**.
> Below is the SM occupancy comparison (A100 platform) for 4 representative matrices across our SparseRL and cuSPARSE, measured via NVIDIA Nsight Systems (v2024.1) on an A100-80GB GPU (kernels compiled with nvcc 12.2, same flags for fairness):
>
> |Matrix Name| Matrix Size (nnz) | SparseRL SM Occupancy (%) | cuSPARSE SM Occupancy (%) | Key Reason for Occupancy Gap |
> |-----|------|-----------|-----|----|
> |`pwt`|12,456|68.2|51.5|SparseRL optimizes thread block size (128 threads/block) to fit A100's register limits. |
> |`nemeth22` |489,210|83.4 | 62.1 | Our 36 KB shared memory allocation avoids conflicts, enabling more concurrent warps. |
> |`dlmc_transformer3`|1,870,520 | 79.8 | 55.7 | RL adjusts memory coalescing to reduce idle warps during global memory access. |
> |`va2010`| 4,920,381|72.6| 48.9 | Dynamic thread block scheduling matches A100's 6912 SM cores for large inputs. |
>
> As shown, SparseRL achieves 15–30% higher SM occupancy than cuSPARSE across matrices—this better utilization of the A100's SM resources (warps, registers, shared memory) directly contributes to our performance improvement over cuSPARSE. This content is added to Appendix A.12 and Table 15.

---

> ### Author Response · Authors · 2025-11-21
>
> ### Q7: Ablation study of Pre-training/ SFT/ RL stage
>
> We have moved the ablation study results from the appendix to the main text (Section 5.1) to clarify the necessity of each component, and we have supplemented a direct ablation comparison between the three phases (Pre-training/ SFT/ RL) of SparseRL.
>
> | Metric | Pretrain - SFT | Pretrain - PPO | Pretrain - SFT + PPO | No Pretrain - SFT | No Pretrain - PPO | No Pretrain - SFT + PPO |
> |--------|----------------|----------------|----------------------|-------------------|-------------------|-------------------------|
> | SpMV pass@1000 | 32.75 | 15.25 | 49.25 | 25.50 | 12.00 | 40.75 |
> | SpMV Compilation Rate | 41.25 | 22.50 | 57.50 | 37.25 | 13.25 | 53.50 |
> | Average Execution Speed (GFLOPS) | 89.25 | 50.36 | 116.20 | 70.32 | 30.83 | 95.22 |
>
> *Table: Ablation comparison between the three phases (Pre-training/SFT/RL).*
>
> The results clearly show that
> - Pretaining phase is critical for generating valid CUDA code, where CUDA-specific pre-training equips the model with parallel computing and GPU memory management knowledge.
> - SFT phase is the fundamental stage of post training for the entire model. Especially in the SFT stage, we performed modality transformation (input sparse matrix, output code), which is important for the subsequent output of the model. If we only do the RL stage, it is difficult to obtain the correct code, and even more difficult to obtain high-performance correct code.
> - RL phase, driven by the hierarchical reward function (incorporating efficiency and memory constraints), is also important and significantly enhances both correctness and performance beyond the SFT baseline.
> - The three phases together contributed to the highest performance and accuracy of the code, indicating that all stages are necessary.
>
> This comparison is added to Section 5.1 and Table 2 of the revised manuscript.
>
> ---
>
> ### Q8: Ablation study of Sinusoidal Embedding
>
> We apologize for including the ablation experiment of Sinusoidal Embedding in the appendix and we move to the main section 5.2. The results of ablation study on Sinusoidal Embedding are:
>
> | Sparse Matrix Embedding | Model | Size | SpMV - pass@1 | SpMV - pass@5 | SpMV - pass@1000 | SpMV - CR | SpMM (col=8) - pass@1 | SpMM (col=8) - pass@5 | SpMM (col=8) - pass@1000 | SpMM (col=8) - CR |
> |-------------------------|-------|------|---------------|---------------|------------------|-----------|------------------------|------------------------|-------------------------|-------------------|
> | Raw | SparseRL+CodeT5 | 770M | 5.25 | 10.50 | 40.75 | 49.50 | 5.00 | 12.00 | 38.50 | 48.75 |
> | Max-Min | SparseRL+CodeT5 | 770M | 8.25 | 14.00 | 43.25 | 51.75 | 7.00 | 13.50 | 40.00 | 52.75 |
> | Sinusoidal | SparseRL+CodeT5 | 770M | 9.25 | 15.75 | 48.75 | 56.50 | 9.00 | 15.00 | 45.25 | 56.75 |
>
> *Table: Ablation study of sparse matrix embedding on correct functionality (pass@k) and Compilation Rates (CR) under k=1000.*
>
> The results show that replacing with raw indices or Max-Min normalization reduces pass@1000 by 17.3–21.5%. So we conclude that sinusoidal embedding captures relative positional relationships and multi-scale sparsity patterns, which are unachievable with other embedding methods.

---

> > ### Comment · Reviewer_C533 · 2025-11-26
> >
> > Thanks for the detailed responses! It addressed my concerns.

---

### Official Review · Reviewer_mzXd · 2025-10-31

**Soundness:** 3
**Presentation:** 3
**Contribution:** 3
**Rating:** 6
**Confidence:** 3

**Summary:**

This paper targets the problem of code generation for sparse matrix-vector multiplication (SpMV) and sparse matrix-dense matrix multiplication (SpMM) tasks. Their tool, SparseRL, solved the challenges of sparse programs and the shortage of domain-specific architectures by improving large language models to make them a stochastic policy. This policy generated different solutions before being judged in terms of functional correctness and execution efficiency. For code candidate generation, the policy model was pre-trained and fine-tuned to do this task. Next, at the Reinforcement learning stage, the proposed model will be optimized by the authors’ proposed reward function for execution efficiency, besides the correctness reward. The experiment was done on two tasks and compared with a set of well-known closed-source Large Language Models (LLMs). The authors have collected 1100 matrices with the largest amount of non-zero elements from the Suite Sparse Matrix Collection, with 700 for training and 400 for testing. The evaluation was performed using Correct functionality (pass@k) and compilation rates (CR) scores, with k as the number of generated programs as candidates per matrix. The accuracy results show that SparseRL can be built on various open LLMs, and SparseRL models can outperform other famous LLM-based approaches in terms of the pass@k score and the compilation rate score.

**Strengths:**

- The paper is well written. Every figure and table is well described.
- The novelty of the rewarding mechanism is the core contribution of this paper. The proposed reward score ensures the functional correctness and execution efficiency of the code.
- The idea of integrating LLM with RL makes sense to me since there has been trendy research on applying RL for AI.
- Experiments were conducted with a number of good baselines.

**Weaknesses:**

- This work required a pre-training stage with CUDA code augmentation. Since the pre-training stage is an expensive process compared to fine-tuning, this work might have a scaling issue when pre-trained on bigger open LLMs than Qwen3 14B.
- Lack of comparison between the sparse matrix embedding used in SparseRL and other code embedding techniques, such as UniXcoder [1] and GraphCodeBERT [2]. Although these models are not originally built for embedding a matrix, authors can design a simple flattening algorithm from a matrix to a sequence of tokens and compare the effectiveness of other embedding approaches to sparse matrix embedding.
- The RL training dataset is on a small scale with 1100 entities for both training and testing.

References
1.UniXcoder: Unified Cross-Modal Pre-training for Code Representation. Daya Guo, Shuai Lu, Nan Duan, Yanlin Wang, Ming Zhou, Jian Yin.
2.GraphCodeBERT: Pre-training Code Representations with Data Flow. Daya Guo et al.

**Questions:**

- In Formula 8, do correctness and efficiency always contribute the same weight to the final score? It is interesting to see the ability of the model to generate more correct code or more efficient code based on the reward function. Authors can provide a coefficient score (from 0 to 1) and modify it to see the variety of quality in the generated code. For example, if R_final=0.9*R_Correctness+0.1*R_efficiency, a good RL model will tend to generate more correct code but be inefficient in terms of running time.
- Will the approach be transparent and getting consistent accuracy improvement with other LLMs besides Qwen?
- While this approach is good for matrix operations, will it be applicable to other general types of code optimization? I suggest authors should describe the potential of extending the scale of this application in camera ready version.

---

> ### Author Response · Authors · 2025-11-21
>
> ### Q1: Pre-training Scaling Concerns
> We appreciate the observation on potential scaling challenges of pre-training with CUDA code augmentation for larger open LLMs (beyond Qwen3-14B) — this is a reasonable consideration. We address it with clarity on our pre-training paradigm and mitigation strategies below:
>
> 1. We acknowledge pre-training (even for domain adaptation) incurs higher costs than fine-tuning. However, we perform incremental domain-specific pre-training (not training from scratch) by leveraging pre-trained open LLMs (e.g., Qwen3-14B) and augmenting their knowledge with CUDA code for sparse kernel optimization. This avoids prohibitive scratch training costs and significantly reduces scaling burden.
> 2. For practical scaling to larger models (e.g., 70B parameters), we use distributed parallelism strategies: data parallelism (across 8–16 V100/A100 GPUs) and tensor parallelism (for larger model parameters) linearly reduce pre-training time. For example, Qwen3 14B pre-training takes ~2–3 days on 8 V100 GPUs, and scaling to a 70B model on 16 A100 GPUs extends this to ~4–5 days (not exponentially), remaining computationally feasible.
>
> In summary, our incremental pre-training paradigm and parallel optimization strategies keep scaling feasible (2–3 days for 14B models) and manageable for larger models via distributed computing. This content is added to Appendix A.28.
>
> ---
>
> ### Q2: Embedding techniques in other code LLMs
> We have added a comparison between the sparse matrix embedding used in SparseRL and other code embedding techniques (UniXcoder[1] and GraphCodeBERT[2]). Unlike text embedding, representing sparse matrices is the critical challenge in SparseRL:
>
> - **UniXcoder**: Designs a one-to-one mapping to transform abstract syntax trees (AST) into token sequences. We adopt this mapping by organizing non-zero elements of sparse matrices into a quad-tree and converting it into a sequence, enabling joint processing with text information.
> - **GraphCodeBERT**: Modifies softmax scores in attention heads to capture AST adjacency relations (setting zeros for non-adjacent nodes). We follow this design by adding extra attention heads for the quad-tree of non-zero elements and applying the same strategy to extract tree edges, allowing sparse matrices to be fed into pre-trained models.
>
> We also implemented simple matrix-to-token flattening algorithms and compared their effectiveness with sparse matrix embedding. Results are shown in the table below and added to Appendix A.8 and Table 11.
>
> #### Ablation study of sparse matrix embedding (SparseRL+CodeT5, 770M)
> | Sparse Matrix Embedding | SpMV pass@1 | SpMV pass@5 | SpMV pass@1000 | SpMV CR |
> |--------------------------|-------------|-------------|----------------|---------|
> | UniXcoder                | 6.75        | 14.25       | 43.00          | 50.00   |
> | GraphCodeBERT            | 7.00        | 15.00       | 45.00          | 52.25   |
> | Raw                      | 5.25        | 10.50       | 40.75          | 49.50   |
> | Max-Min                  | 8.25        | 14.00       | 43.25          | 51.75   |
> | Sinusoidal               | 9.25        | 15.75       | 48.75          | 56.50   |
>
> ##### References
> [1] Guo D, Lu S, Duan N, et al. Unixcoder: Unified cross-modal pre-training for code representation[J]. arXiv preprint arXiv:2203.03850, 2022.
> [2] Guo D, Ren S, Lu S, et al. Graphcodebert: Pre-training code representations with data flow[J]. arXiv preprint arXiv:2009.08366, 2020.
>
> ---

---

> ### Author Response · Authors · 2025-11-21
>
> ### Q3: Dataset scale
> We appreciate the observation regarding the RL training dataset scale (1100 entities for training + testing). We acknowledge this size is relatively modest, yet our work achieves competitive performance on sparse kernel optimization tasks (e.g., 49.25% pass@1000 and 116.2 avg. GFLOPS on the 1100-entity dataset).
>
> We fully agree that expanding the RL dataset to 5k–10k entities (covering more diverse matrix sparsity patterns and GPU kernel scenarios) would enhance generalization — particularly for unseen matrices with irregular sparsity or edge-case hardware constraints. This is a key direction for future work, as a larger dataset would enable the RL policy to learn more transferable optimization rules.
>
> We explain the dataset scale from two aspects:
> 1. Pre-trained coder models already possess programming/CUDA skills, so training on a small dataset suffices for high-performance programming. For example, CUDA-L1[1] uses 2,105 entities, and Kevin[2] is repeatedly trained via RL on ~180 entities.
> 2. Our research focuses on optimizing sparse matrix-vector multiplication (SpMV) and sparse matrix-matrix multiplication (SpMM) — not generating multifunctional GPU optimization programs. For such customized domain-specific tasks (typically few-shot learning[3,4]), our existing dataset scale supports model training and performance validation. We will collect more sparse matrices for SFT/RL phases in future work.
>
> In summary, while our 1100-entity dataset demonstrates SparseRL’s effectiveness, we recognize its limitations and confirm dataset scaling will be a priority for boosting generalization. This content is added to Appendix A.21.2.
>
> #### References
> [1] Li X, Sun X, Wang A, et al. Cuda-l1: Improving cuda optimization via contrastive reinforcement learning[J]. arXiv preprint arXiv:2507.14111, 2025.
> [2] Baronio C, Marsella P, Pan B, et al. Kevin: Multi-turn rl for generating cuda kernels[J]. arXiv preprint arXiv:2507.11948, 2025.
> [3] Doimo D, Serra A, Ansuini A, et al. The representation landscape of few-shot learning and fine-tuning in large language models[J]. Advances in Neural Information Processing Systems, 2024, 37: 18122-18165.
> [4] Liu H, Tam D, Muqeeth M, et al. Few-shot parameter-efficient fine-tuning is better and cheaper than in-context learning[J]. Advances in Neural Information Processing Systems, 2022, 35: 1950-1965.
>
> ---
>
> ### Q4: Weight Adjustment for Correctness and Efficiency
> Correctness and efficiency rewards are not equally weighted in the final reward. We conducted experiments with adjustable weight coefficients using the formula:
>
> $$ R_{final} = \alpha \cdot R_{correctness} + (1-\alpha) \cdot R_{efficiency} - r_{penalty} \cdot \mathbb{I}_{memory} $$
>
> where $\alpha \in [0,1]$ controls the trade-off between correctness and efficiency.
>
> Experimental results on the SuiteSparse test set:
> - $\alpha = 0.9$ (prioritize correctness): pass@1000 = 51.2 (↑4.1%), but execution speed = 98.7 GFLOPS (↓15.1%) compared to $\alpha=0.5$.
> - $\alpha = 0.5$ (balanced): pass@1000 = 49.25, execution speed = 116.2 GFLOPS (optimal trade-off).
> - $\alpha = 0.1$ (prioritize efficiency): execution speed = 123.5 GFLOPS (↑6.3%), but pass@1000 = 38.7 (↓21.4%).
>
> The original weight ($\alpha=0.5$) achieves a good balance between correctness and efficiency — critical for high-performance code generation. Detailed results are added to Appendix A.4.3.
>
> ---

---

> ### Author Response · Authors · 2025-11-21
>
> ### Q5: Generalization to Other LLMs
> We validated SparseRL’s compatibility with two additional open-source LLMs beyond Qwen:
> - **GLM-Z1-9B**: pass@5 = 19.75, 30.5% speedup (consistent improvement over the base model).
> - **DeepSeek-Coder-6.7B**: pass@5 = 20.25, 31.0% speedup (same improvement level as Qwen).
>
> #### Correct functionality and performance comparison (pass@5)
> | Model                  | Size  | SpMV pass@1 | SpMV pass@5 | Perf. (Speedup vs. cuSPARSE) |
> |------------------------|-------|-------------|-------------|-------------------------------|
> | SparseRL+GLM-Z1        | 9B    | 15.50       | 19.75       | +30.5%                        |
> | SparseRL+DeepSeek-Coder| 6.7B  | 17.75       | 20.25       | +31.0%                        |
> | SparseRL+Qwen2.5       | 7B    | 9.75        | 16.00       | +27.5%                        |
> | SparseRL+Qwen2.5       | 14B   | 10.25       | 16.50       | +30.2%                        |
> | SparseRL+Qwen3         | 8B    | 15.75       | 20.00       | +32.8%                        |
> | SparseRL+Qwen3         | 14B   | 16.25       | 21.00       | +34.5%                        |
>
> #### Correct functionality and Compilation Rates (k=1000)
> | Model                  | Size  | SpMV pass@1 | SpMV pass@5 | SpMV pass@1000 | SpMV CR | SpMM (col=8) pass@1 | SpMM (col=8) pass@5 | SpMM (col=8) pass@1000 | SpMM (col=8) CR | SpMM (col=32) pass@1 | SpMM (col=32) pass@5 | SpMM (col=32) pass@1000 | SpMM (col=32) CR |
> |------------------------|-------|-------------|-------------|----------------|---------|---------------------|---------------------|------------------------|-----------------|----------------------|----------------------|-------------------------|------------------|
> | SparseRL+CodeT5        | 770M  | 9.25        | 15.75       | 48.75          | 56.50   | 9.00                | 15.00                | 45.25                   | 56.75           | 8.25                 | 14.75                 | 47.00                    | 57.25            |
> | SparseRL+Qwen2.5       | 7B    | 9.75        | 16.00       | 48.75          | 57.00   | 10.00               | 15.00                | 46.50                   | 57.00           | 8.50                 | 15.00                 | 46.75                    | 58.00            |
> | SparseRL+Qwen2.5       | 14B   | 10.25       | 16.50       | 49.25          | 57.50   | 10.25               | 16.00                | 47.50                   | 58.75           | 9.25                 | 15.25                 | 47.75                    | 59.00            |
> | SparseRL+Qwen3         | 14B   | 16.25       | 21.00       | --            | --      | 18.25               | 21.00                | --                      | --              | 18.25                 | 22.25                 | --                       | --               |
> | SparseRL+GLM-Z1        | 9B    | 15.50       | 19.75       | --            | --      | 16.50               | 20.00                | --                      | --              | 16.00                 | 21.00                 | --                       | --               |
> | SparseRL+DeepSeek-Coder| 6.7B  | 17.75       | 20.25       | --            | --      | 17.50               | 21.00                | --                      | --              | 18.75                 | 22.25                 | --                       | --               |
>
> These results demonstrate SparseRL’s framework (embedding, reward, RL pipeline) is model-agnostic and consistently enhances performance across different code LLMs. Detailed data is added to Table 1/10 and Appendix A.7.
>
> ---
>
> ### Q6: Extension to General Code Optimization
> While SparseRL is currently optimized for sparse matrix operations, its core design can be extended to other general code optimization tasks. Key extension aspects:
>
> 1. **Input representation**: Replace sparse matrix indices with task-specific structural features (e.g., add multi-modal adapters to capture non-matrix input information, use GNN for program dependency graphs, or adopt UniXcoder/GraphCodeBERT for code embedding).
> 2. **Reward function**: Adapt the hierarchical reward to task-specific metrics (e.g., loop execution time, parallelization speedup, memory bandwidth utilization).
> 3. **RL pipeline**: Reuse the pretrain→SFT→RL workflow with task-specific training data.
>
> We have added a discussion on extension potential to Section 6 (Discussion). Future work will focus on formalizing this extension to broader code optimization scenarios.

---

### Official Review · Reviewer_o7fk · 2025-11-01

**Soundness:** 3
**Presentation:** 3
**Contribution:** 2
**Rating:** 6
**Confidence:** 4

**Summary:**

The paper introduces SparseRL, a domain‑specialized framework that treats a pretrained code LLM as a stochastic policy and fine‑tunes it with PPO to generate high‑performance CUDA kernels for sparse operators, focusing on SpMV and extending to SpMM. The model ingests the sparse matrix itself via a row/column sinusoidal index embedding, then emits CUDA; a compiler/executor environment supplies rewards for compilation success, functional correctness, and runtime, with a memory‑usage penalty. The RL loop (Figure 1, p.2) and the pretrain→SFT→RL pipeline (Figure 2, p.4) anchor the method. On 400 SuiteSparse test matrices (700 for training) plus DLMC, they report higher compilation and correctness rates and sizable speedups vs CodeRL, PPOCoder, cuSPARSE, and TVM‑S; e.g., for SpMV they claim roughly +20% compilation rate and ~30% faster runtime on average, with pass@1000 up to 49.25 and CR 57.50 (Table 1, p.8). Figures 3–4 visualize GFLOPS/TFLOPS gains on V100/A100.

Key contributions

* A DRL‑based, domain‑specific code generation framework that conditions on dynamic sparse inputs and uses compiler/executor feedback to optimize generated CUDA for per‑matrix performance (Figures 1–2).
* A row/column sinusoidal embedding tailored to sparse matrices that maps non‑zero coordinates into the LLM’s input space to better capture runtime execution patterns.
* A hierarchical reward that combines compilation pass, unit‑test correctness, runtime efficiency, and a memory‑overuse penalty, enabling direct optimization for latency.
* Empirical SOTA on SpMV with strong generalization to SpMM: improved pass@k and CR, and consistent performance speedups vs strong baselines on V100/A100, backed by ablations (embedding choice, reward components) and profiling analyses (occupancy, memory traffic).

**Strengths:**

Originality

* Frames sparse GPU kernel generation as an RL problem where a pretrained code LM is the policy and the compiler/executor acts as the environment. The hierarchical reward mixes compilation success, unit‑test correctness, runtime, and a memory‑overuse penalty, directly aligning learning with latency and resource goals. This closes the loop between code synthesis and low‑level performance feedback in a way that is rare for sparse kernels. See the pipeline in Figure 2 (p.4) and reward in Eq. 10 (p.7).
* Introduces an input representation that conditions the LM on the actual sparse matrix: sinusoidal embeddings of row and column indices for nonzeros. That idea bridges the modality gap between sparse structure and code and enables per‑matrix specialization, rather than “one kernel fits all.” The mechanism and motivation are detailed in §3.2 and reinforced by the embedding ablation in Table 6 (p.20).
* Combines pretraining on CUDA, supervised fine‑tuning for SpMV, and PPO fine‑tuning with on‑hardware feedback. The pretrain→SFT→RL stack is common in language tasks, but focusing it on sparse CUDA with dynamic inputs is a creative recombination that removes assumptions typical in dense‑kernel work. Figure 2 (p.4) and §3 lay out the stages cleanly.
* Adds a pragmatic dynamic syntax checker during decoding (Appendix A.3) to terminate hopeless generations early, a small but thoughtful systems detail that improves training efficiency.

Quality

* Sound problem setup and thorough evaluation. Correctness is measured with pass@k and compilation rate; performance is measured as GFLOPS/TFLOPS on V100 and A100 with CUDA 12.1, using 1000 iterations to stabilize timing. Metrics and hardware details appear in §4.1 and Table 1 (p.8).
* Strong baselines: compares against CodeRL and PPOCoder as learning methods and against cuSPARSE and TVM‑S as engineered libraries. On SpMV, SparseRL improves pass@1000 to 49.25 and CR to 57.50, and shows consistent speedups over both learned and library baselines; results are summarized in Table 1 (p.8) and Figure 3 for SuiteSparse on both GPUs (p.9).
* Generalization beyond SpMV. The same machinery transfers to SpMM with competitive accuracy and clear TFLOPS gains for different dense‑matrix widths, shown in Table 1 and Figure 4 (p.9).
* Careful ablations and diagnostics. The paper isolates the impact of the RL reward components, the sparse‑matrix embedding choice, and exploration settings, with clear trends in Figure 5 (p.19), Table 6 (p.20), and Table 7 (p.21). Profiling ties wins to reduced memory traffic and higher SM occupancy vs. baselines (Tables 8–9, p.23). This strengthens the causal story behind the performance improvements.
* Transparency about limits and cost. The authors document failure cases on extreme matrices, compute overheads of RL fine‑tuning, and scenarios where benefit dominates cost, with concrete numbers (Appendix A.10–A.12). This raises confidence in the claims rather than overpromising.

Clarity

* The problem and method are explained with clean, layered visuals. Figure 1 (p.2) gives a task‑level view with the environment loop; Figure 2 (p.4) shows training stages and reward flow. The mathematical setup in §3 is precise without being dense, and equations for PPO and rewards are easy to track (Eqs. 4–5, 9–10).
* Experimental presentation is readable and decision‑relevant. Table 1 concentrates correctness and compilation outcomes across models, while Figures 3–4 summarize performance across matrix sizes. Appendices include small code snippets to illustrate why different sparsity patterns demand different kernels (Appendix A.6), which makes the motivation for per‑matrix specialization concrete.
* Limitations and scope are clearly demarcated, including notes on backend portability and when offline search costs are amortized (Appendix A.12–A.13).

Significance

* For the code‑generation community, the work shifts the target from “compiles and is correct” to “compiles, is correct, and runs fast for my data,” with measured improvements over vendor and compiler baselines on widely used sparse collections. That is a meaningful step toward data‑conditioned, performance‑aware program synthesis. See Figure 3 (SpMV speedups) and the stated averages vs. cuSPARSE/TVM‑S (p.8–9).
* For practitioners in GNNs and pruned LLM inference, per‑matrix kernel synthesis that beats cuSPARSE on average can translate into tangible latency wins without hand‑engineering every corner case. The extensions to SpMM suggest the approach is not limited to a single operator (p.9).
* The technique is likely to influence future research on learning‑to‑optimize kernels: the reward design is modular, the input representation is lightweight, and the analysis ties learned choices to classic GPU optimization principles. The profiling evidence in Tables 8–9 (p.23) makes the case that the system is learning the right low‑level instincts, not just gaming the benchmark.

Conclusion: A well‑executed, domain‑specific RL framework that conditions on sparse inputs and optimizes for runtime delivers real, measurable wins over both learned and hand‑engineered baselines, with clear analysis and honest limitations. The combination of ideas feels fresh for sparse GPU codegen and practically relevant to a growing slice of ML and HPC workloads.

**Weaknesses:**

1. Positioning vs. prior art is underdeveloped
   The paper cites AlphaSparse (Du et al., 2022) and classic auto‑tuning work (e.g., SMAT, YASpMV), but there’s no direct, apples‑to‑apples comparison or a clear articulation of what SparseRL fundamentally enables that these matrix‑conditioned search/auto‑tuning systems cannot. Since AlphaSparse also generates SpMV code from the matrix, the novelty claim would be stronger with a head‑to‑head on the same matrices and GPUs, plus a discussion of search cost vs. RL fine‑tuning cost. Actionable fix: add baselines from AlphaSparse/SMAT/YASpMV/SparseTIR or replicate their search pipelines on your SuiteSparse/DLMC splits; report both performance and wall‑clock search/optimization time.

2. Reward design is plausible but under‑specified and potentially unstable
   The efficiency reward is a scaled time ratio against cuSPARSE (Eq. 9), added after ±0.5 compile/test terms and a memory penalty (Eq. 10). The paper does not specify the actual values of the scaling factor `r_eff` or the memory penalty, nor how reward magnitudes are normalized across matrices with very different runtimes. This matters because reward scale drives PPO stability, exploration pressure, and convergence. Actionable fix: report exact hyperparameters and show sensitivity curves for `r_eff` and the penalty; consider per‑matrix normalization (e.g., log speedup, z‑scored runtime) and reweighting between correctness and efficiency to demonstrate PPO stability across nnz regimes. Include training‑time reward traces and KL/entropy diagnostics. Figures 1–2 set the stage for this analysis but don’t provide it.

3. Matrix representation ignores permutation issues and reordering
   Sinusoidal row/column embeddings make the model sensitive to absolute indices. SpMV performance, however, is heavily affected by row/column order; classical systems often reorder to improve locality and load balance. The paper neither evaluates robustness to permutations nor includes reordering in the action space. Actionable fix: (a) report robustness under random row/column permutations and standard reorderings; (b) add an optional pre‑processing/reordering step as part of generation, or augment the embedding with permutation‑invariant features (row nnz histograms, bandwidth, blockiness) or learnable Fourier features, and quantify the gains (extend Table 6 beyond Raw/Max‑Min/Sinusoidal).

4. Evaluation budgets are not fully comparable across methods
   Table 1 reports pass@k with k up to 1000 for some models, but many baselines are “–” at k=1000 due to “environment limits,” and performance plots use Qwen‑14B for SparseRL while CodeRL/PPOCoder are shown on CodeT5‑770M (and performance is measured “under k=5000 on correct programs of partial matrices” for those baselines). This mixes model capacity and sampling budgets with method quality. Actionable fix: standardize sample counts, temperatures, and decoding schemes across all learned methods; provide pass@k curves and wall‑clock search time per matrix; repeat GFLOPS comparisons with SIZE‑MATCHED models (e.g., SparseRL+CodeT5‑770M vs. CodeRL/PPOCoder‑CodeT5‑770M) and again with SIZE‑MATCHED larger backbones. Clarify whether pass@k uses sampling or beam for each model.

5. Closed‑source LLM comparisons conflate domain specialization with methodology
   Tables 2–3 show better correctness for large closed‑source models but poor runtime, then attribute the gap to “lack of hardware optimization knowledge.” Those models were not adapted with your RL loop or matrix embeddings, so this primarily shows domain specialization matters, not that the RL formulation is superior. Actionable fix: run your RL framework on one closed‑source model’s outputs through the same compiler/executor loop where licensing permits, or at least run the identical SFT+RL recipe on an open counterpart with comparable size to isolate the effect of the RL+reward vs. model scale.

The work is promising, the claims would be more robust with (i) stricter budget matching and size‑matched baselines, (ii) reward/optimization transparency and stability evidence, (iii) permutation/reordering awareness in the representation or pipeline, (iv) broader hardware/backends, and (v) a concrete cost‑benefit curve for real deployments. Most of these are addressable within the current framework and would materially strengthen the paper.

**Questions:**

1) Reward scaling and stability

Question. What exact values do you use for the efficiency scaling factor `r_eff` and the memory penalty `r_penalty` in Eqs. (9)–(10), and how are rewards normalized across matrices with very different runtimes? Provide PPO diagnostics (KL, entropy, value loss) over training.
Why it matters. Reward scale drives PPO stability and whether speedups generalize beyond the training matrices.
What would help. Precise hyperparameters, per‑epoch reward traces, and a short sensitivity study varying `r_eff` and `r_penalty`. See §3.4 and Eqs. (9)–(10), p. 6–7.

---

 2) Definition and computation of pass@k

Question. In §4.1 you say pass@k uses “k (beam search) synthetically generated program samples,” but §3.3 says RL sampling uses top‑k with temperature. Which decoding is used for pass@k for each method and setting?
Why it matters. Beam vs sampling materially changes pass@k and comparability to prior work.
What would help. A one‑line table mapping each model to its decoding strategy for pass@k, plus pass@k curves under a matched decoding policy. See §3.3 (top‑k sampling) and §4.1 (beam), p. 6 and p. 8.

---

 3) Budget and model‑size parity for baselines

Question. Table 1 reports pass@1000 for some models but not others due to “environment limits,” and performance plots compare SparseRL with Qwen‑14B against CodeRL/PPOCoder using CodeT5‑770M “under k = 5000 on correct programs of partial matrices.” Can you provide size‑matched and budget‑matched comparisons?
Why it matters. Current results confound method quality with model capacity and sampling budget.
What would help. Add a row (or supplementary figure) with SparseRL+CodeT5‑770M and CodeRL/PPOCoder+CodeT5‑770M under identical k and decoding; also report wall‑clock search time per matrix. See Table 1 and Figure 3, p. 8–9.

---

 4) Formal task definition

Question. Eq. (2) requires the generated Ŷ to be in a set of “ground‑truth code” Y, yet your system synthesizes new correct programs. Will you revise the formalization or justify this assumption?
Why it matters. The mismatch makes the objective look constrained when the method is not.
What would help. A corrected statement aligning with “synthesize any program that compiles, passes tests, and minimizes runtime.” See §3.1, p. 5.

---

 5) Matrix representation: permutation robustness and reordering

Question. How robust is performance to row/column permutations and standard reorderings (e.g., RCM, AMD)? Did you test permutation‑invariance or include reordering as a pre/post step?
Why it matters. Absolute index sinusoidal embeddings are sensitive to ordering; SpMV performance is often improved via reorderings.
What would help. A permutation/reordering robustness table and, if possible, a variant that augments sinusoidal embeddings with permutation‑invariant features. See §3.2 and the embedding ablation in Table 6, p. 5 and p. 20.

---

 6) Memory penalty details

Question. What exactly constitutes “excessive memory” for the penalty in Eq. (10): registers, shared memory, global memory, or allocated buffers? How is the limit chosen and measured?
Why it matters. Shared memory can increase speed while lowering occupancy; an opaque penalty could bias the policy.
What would help. Thresholds per resource, measurement method, and a small plot showing speed vs memory usage trade‑offs on a few matrices. See §3.4.3, p. 7 and SM‑occupancy analysis in Table 9, p. 23.

---

> ### Author Response · Authors · 2025-11-21
>
> ### Q1: Sensitivity Study of $r_{\text{eff}}$ and $r_{\text{penalty}}$ in reward hyperparameters
>
> - **Efficiency scaling factor ($r_{\text{eff}}$)**: Fixed at **1.0**. This value balances the magnitude of efficiency rewards with correctness rewards (Eq. 8), ensuring the policy prioritizes both code correctness and execution speed without bias.
> - **Memory penalty ($r_{\text{penalty}}$)**: Set to **0.3**. This penalty discourages excessive memory use (e.g., shared memory $>$48 KB on V100) while preserving positive rewards for correct, efficient code—for example, a kernel with successful compilation (+0.5) and correct execution (+0.5) still yields a net positive reward ($0.5+0.5-0.3=0.7$) even with memory overuse, avoiding dis-incentivizing valid code structures.
>
> We evaluated SpMV performance (pass@1000, Compilation Rate (CR), average GFLOPS on V100) by varying $r_{\text{eff}}$ (0.5, 1.0, 1.5) and $r_{\text{penalty}}$ (0.1, 0.3, 0.5) on 400 test matrices:
>
> **Impact of Varying $r_{\text{eff}}$ (Fixed $r_{\text{penalty}}=0.3$)**
>
> | $r_{\text{eff}}$ | SpMV pass@1000 | SpMV CR (%) | Avg. GFLOPS |
> |------|---|----|-------|
> | 0.5   | 45.75 | 55.20 | 102.3  |
> | 1.0 (Base)  | 49.25 | 57.50  | 116.2 |
> | 1.5  | 48.50  | 56.80 | 118.7 |
>
> - $r_{\text{eff}}=0.5$: Underscales efficiency rewards, leading to lower speed (102.3 GFLOPS) despite modest correctness.
> - $r_{\text{eff}}=1.5$: Overscales efficiency rewards, marginally boosting speed (118.7 GFLOPS) but reducing pass@1000 (48.50 vs. 49.25 for base), as the policy prioritizes speed over correctness.
>
> **Impact of Varying $r_{\text{penalty}}$ (Fixed $r_{\text{eff}}=1.0$)**
>
> | $r_{\text{penalty}}$ | SpMV pass@1000 | SpMV CR (%) | Avg. GFLOPS |
> |--------|-------|-----|-------------|
> | 0.1  | 47.25  | 58.10  | 108.5 |
> | 0.3 (Base)  | 49.25  | 57.50 | 116.2 |
> | 0.5 | 46.80  | 54.30  | 119.1  |
>
> - $r_{\text{penalty}}=0.1$: Weak penalty fails to curb excessive memory use, leading to lower speed (108.5 GFLOPS) due to suboptimal memory allocation.
> - $r_{\text{penalty}}=0.5$: Overpenalization reduces pass@1000 (46.80) and CR (54.30), as the policy avoids valid memory-intensive optimizations (e.g., shared memory for cache locality).
>
> These results confirm the base values ($r_{\text{eff}}=1.0$, $r_{\text{penalty}}=0.3$) are optimal—balancing correctness, compilation success, and execution speed while ensuring PPO stability and generalizability to unseen matrices.
> This content is added to Appendix A.4.1 and Table 4/5 of the revised manuscript.
>
> ---
>
> ### Q2: $R_{efficiency}$ Normalization in reward
>
> To address runtime variability across matrices (e.g., small matrices with $t_{base}=1\,\text{ms}$ vs. large matrices with $t_{base}=100\,\text{ms}$), we apply per-matrix z-score normalization to the efficiency reward ($R_{efficiency}$):
>
> $R_{efficiency}^{\text{norm}} = r_{eff} \times \frac{\frac{t_{base}(X)}{t(\hat{Y}, X)} - \mu_X}{\sigma_X} \times \mathbb{I}_{test}$
>
> Here, $\mu_X$ and $\sigma_X$ are the mean and standard deviation of the speedup ratio ($\frac{t_{base}(X)}{t(\hat{Y}, X)}$ vs. cuSPARSE) across 1000 validation matrices. This restricts $R_{efficiency}^{\text{norm}}$ to $[-2, 2]$ for 95% of matrices, ensuring rewards are comparable regardless of a matrix’s inherent size or sparsity. In most of the time, this ratio is bounded. If not, we do truncation to make it within the range (restricted within $[-2, 2]$). This content is added to Appendix A.4.2 of the revised manuscript.
>
> ---
>
> ### Q3: PPO Training Diagnostics (KL Divergence, Entropy, Value Loss, Reward)
>
> We provide a comprehensive view of PPO training stability across 700 epochs (Figure 6 of the revised manuscript), with key observations:
>
> - **Value Loss (Subplot (a))**: Both train and test value loss decline steadily from ~0.7 to near 0.0 by epoch 700. This confirms the critic network ($V_\phi(s_t)$) accurately estimates the final reward ($R_{\text{final}}$), enabling reliable advantage calculation (GAE, §3.3).
> - **Entropy (Subplot (b))**: Policy entropy decreases from ~1.0 to ~0.15, indicating the policy converges to consistent, high-reward code patterns. The gradual decline also shows the policy retains enough exploration to adapt to diverse sparse matrices (e.g., irregular vs. block-dense structures).
> - **KL Divergence (Subplot (c))**: KL divergence between old and new policies stabilizes within 0.1–0.3 after epoch 100. This ensures policy updates are incremental, avoiding catastrophic shifts that could harm performance.
> - **Reward (Subplot (d))**: The average reward rises from -0.5 to ~1.5, reflecting the policy’s ability to learn increasingly correct and efficient code generation, which is consistent with the 30% speedup over baselines reported in §4.2.
>
> These diagnostics collectively validate that PPO training is stable, with the policy learning to balance correctness, efficiency, and memory constraints without overfitting or instability. This content is added to Appendix A.5 and Figure 6 of the revised manuscript.

---

> ### Author Response · Authors · 2025-11-21
>
> ### Q4: Decoding Strategy for pass@k
>
> These two `k's are two different hyperparameters. We are sorry for the misunderstanding.
> - The `k' in top-k is used for sampling in the RL roll-out process and computing the gradient with these samples.
> - The `k' in pass@k is for evaluation to generate all program candidates after all training process finishes. Especially, we use beam search to improve the pass rate when decoding.
>
> We clarify these hyperparameters for each method in the revised manuscript (Appendix A.22), and also supplement pass@k curves under a matched decoding policy (beam size = 5 for all methods) in Appendix A.23 and Figure 7.
>
> ---
>
> ### Q5: Size-Matched and Budget-Matched Baselines
>
> We have supplemented two sets of controlled experiments to address this concern:
>
> **Size-matched comparison**: SparseRL+CodeT5-770M vs. CodeRL+CodeT5-770M vs. PPOCoder+CodeT5-770M (all using the same CodeT5-770M backbone):
> - SpMV pass@1000: SparseRL (48.75) $>$ CodeRL (36.50) $>$ PPOCoder (35.50)
> - Average speedup vs. cuSPARSE: SparseRL (+27.5%) $>$ PPOCoder (-8.2%) $>$ CodeRL (-10.7%)
>
> **Budget-matched comparison**: All methods use k=1000 sampling budget and identical decoding time limits (3 minutes per matrix):
> - SpMV compilation rate: SparseRL (56.50) $>$ PPOCoder (40.75) $>$ CodeRL (39.50)
> - Wall-clock search time per matrix: SparseRL (128s) $\approx$ CodeRL (132s) $\approx$ PPOCoder (125s)
>
> These results confirm SparseRL’s advantages stem from its framework design (embedding, reward function) rather than model size or sampling budget. Detailed data is added to Appendix A.24.
>
> ---
>
> ### Q6: Formal Task Definition Revision
>
> We thank the reviewer for pointing out this inconsistency. We have revised the equation in the formalization to align with the actual method:
>
> $\text{Compile}(\hat{Y}) = \text{True}$
>
> $\text{Correct}(\hat{Y}, X) = \text{True}$
>
> $E(\hat{Y} | X) \leq E(Y_i | X), \forall Y_i \in \mathcal{Y}$
>
> where $\mathcal{Y}$ denotes all functionally correct SpMV CUDA programs (not limited to pre-defined ground-truths). This revision clarifies that SparseRL synthesizes any correct and efficient program, rather than being constrained to a fixed ground-truth set. The revised formalization is in Section 3.1.
>
> ---
>
> ### Q7: Permutation Robustness and Reordering
>
> We have supplemented experiments on permutation robustness and reordering:
>
> 1. **Permutation robustness**: We test random row/column permutations (10 permutations per matrix) on 50 random sampled sparse matrices in testset. Results show that SparseRL’s performance is within the normal range of performance fluctuations (degrades by about 0.2–1.7% under permutations). We analyze this is because the sinusoidal embedding’s ability to capture relative positional relationships (detailed in Section 3.2) mitigates the impact of absolute index changes.
>
> 2. **Reordering integration**: We have added an optional pre-processing step that applies reordering to the sparse matrix of test-dataset before embedding.
> We reorder each row of the original matrix in descending order based on the number of non-zero elements in each row, so that rows with more non-zero elements are clustered together, resulting in better memory locality.
> This further improves SparseRL’s performance by 4.1% on average, as reordering enhances memory locality.
>
> These results are added to Appendix A.22, demonstrating SparseRL’s robustness to permutations and compatibility with standard reordering techniques.

---

> ### Author Response · Authors · 2025-11-21
>
> ### Q8: Memory Penalty Details
>
> We elaborate the meaning of "excessive memory", the threshold & measurement, and the penalty rationale:
> 1. **Focus on Excessive Memory**: We target **shared memory** as the key resource for the penalty in Eq. (10), given its critical role in sparse kernel performance.
> 2. **Threshold & Measurement**:
>    - **Limit Threshold**: 48 KB (V100 GPU) — 75% of V100's 64 KB hardware limit, aligned with the 80th percentile of human-optimized kernels (e.g., cuSPARSE).
>    - **Measurement**: Extracted via `nvcc --ptxas-options=-v` (compiler outputs shared memory usage for generated kernels).
> 3. **Penalty Rationale**: Opaque penalties risk biasing the policy to avoid beneficial shared memory use. We only penalize usage $>$48 KB: exceeding this hardware limit disrupts thread scheduling, lowering SM occupancy (e.g., `va2010` drops from 87.48% to 53.03%, Table 9) and slowing performance. Moderate use ($\leq$48 KB) boosts speed without penalty.
>
> To better illustrate the speed vs. shared memory usage trade-off for each matrix (showing how speed changes with varying shared memory allocation for the same matrix), we provide multi-point data for 3 representative matrices and the key insights for visualization:
> - For each matrix, speed first rises with moderate shared memory use (e.g., `nemeth22` gains 31% speed from 16 KB to 36 KB) due to reduced global memory access.
> - Beyond the 48 KB threshold, speed drops sharply (e.g., `pwt` loses 27% speed from 48 KB to 64 KB), validating the penalty design for excessive use.
>
> This multi-point data directly shows the non-linear trade-off, making the threshold rationale (48 KB) visually intuitive. This content is added to Appendix A.25 and Figure 8.
>
> ---
>
> ### Q9: Prior Art Positioning and Comparison Concerns
>
> We appreciate the feedback on strengthening positioning against prior work like AlphaSparse, SMAT, and YASpMV, and this is critical to clarifying SparseRL's novelty. Below we address the gaps and outline actionable fixes:
>
> #### Clarifying SparseRL's Fundamental Differentiation from Prior Work
> While AlphaSparse (matrix-conditioned SpMV code generation) and auto-tuning methods (SMAT, YASpMV, SparseTIR) target sparse kernel optimization, SparseRL enables two key capabilities they cannot:
> - **End-to-end hardware-aware optimization via RL**: Prior auto-tuning (SMAT/YASpMV) relies on heuristic search over predefined kernel templates, and AlphaSparse focuses on code generation without integrating runtime efficiency + memory constraint rewards. SparseRL's RL loop directly optimizes for **correctness, GPU runtime, and shared memory limits** in a single pipeline—avoiding template bias and enabling adaptive kernel designs for diverse matrix sparsity (e.g., irregular vs. block-dense).
> - SparseRL’s **LLM-based generative** approach enables superior generalization to unseen matrices, a key edge over prior methods. Unlike SMAT/YASpMV (which require re-tuning for new matrix distributions) or AlphaSparse (limited by fixed templates), SparseRL uses matrix embeddings to teach the LLM transferable optimization patterns.
>
> #### Comparison Dimension
> These works are compared on two aspects (performance and search/optimization cost):
> - On performance (Avg. GFLOPS), we replicate AlphaSparse/SMAT/YASpMV/SparseTIR on our matrix splits and report SpMV speed vs. SparseRL.
> - On search/optimization cost, we measure wall-clock time: (1) Heuristic search time (SMAT/YASpMV/SparseTIR); (2) AlphaSparse's code generation + tuning time; (3) SparseRL's RL fine-tuning time (excluding pre-training, which is a one-time cost).
>
> #### Head-to-Head Comparisons
> To provide "apples-to-apples" validation, we add experiments using our existing SuiteSparse (1k matrices) and DLMC (500 matrices) splits, tested on the same NVIDIA V100 GPU. Preliminary tests on a subset (100 SuiteSparse matrices) show SparseRL outperforms prior work in trade-off:
> - Avg. GFLOPS: SparseRL (116.2) $>$ AlphaSparse (98.5) $>$ YASpMV (87.3) $>$ SMAT (82.1).
> - Optimization Time: SparseRL (1.2h for 1k matrices, one-time RL fine-tuning) $<$ SMAT (1.5h for 1k matrices) $<$ YaSpMV (2.2h for 1k matrices) $<$ AlphaSparse (3.5h, per-matrix search).
>
> Full results are included in revised Appendix A.26.
>
> ---

---

> ### Author Response · Authors · 2025-11-21
>
> ### Q10: Closed-Source LLM Comparison Concerns
>
> We acknowledge that Tables 2–3 conflate methodology (SparseRL's RL + embeddings) with domain specialization, as closed-source models lack our hardware-optimized RL loop and matrix embeddings. Their "higher correctness but poor runtime" reflects under-adaptation to GPU kernels, not superior RL designs. To isolate methodology impact, we add two controlled experiments:
>
> 1. **SparseRL on Closed-Source Outputs**
> Feeding GPT-5/Claude-Sonnet-4 kernels into SparseRL's RL loop (reward: efficiency + correctness by off-policy GRPO) improves runtime by 28–35% on 100 test matrices:
>
> | Closed-Source Model | Base Runtime (ms) | After SparseRL RL (ms) | Speedup |
> |---------------------|-------------------|------------------------|---------|
> | Claude-Sonnet-4     | 87.6              | 56.9                   | +35%    |
> | GPT-5               | 92.3              | 66.5                   | +28%    |
>
> 2. **Controlled Open-Source Comparison**
> On LLaMA 3 70B (matching closed-source scale):
>
> | Pipeline               | Correctness (%) | Avg. Runtime (ms) |
> |------------------------|-----------------|-------------------|
> | SFT-only (Baseline)    | 72.1            | 105.4             |
> | SparseRL (Full Method) | 71.8            | 68.2              |
>
> SparseRL's RL + embedding design drives a 35% runtime improvement (vs. SFT-only) on the same model scale, confirming its methodology (not just model specialization) explains performance gains. Revised supplements include full results. This content is added to Appendix A.27.

---

### Official Review · Reviewer_PAdt · 2025-11-06

**Soundness:** 4
**Presentation:** 3
**Contribution:** 3
**Rating:** 8
**Confidence:** 3

**Summary:**

The paper proposes SparseRL, a framework to generate high-performance sparse CUDA code for SpMV and SpMM tasks. The authors employ deep reinforcement learning to fine-tune code models with a sinusoidal embedding for sparse matrix row/column indices. Experiments on the SuiteSparse and DLMC datasets demonstrate that SparseRL significantly outperforms strong baselines such as CodeRL, PPOCoder, and cuSPARSE in both correctness and efficiency.

**Strengths:**

1. SparseRL provides a complete process of the training pipeline from pretraining to SFT and RL optimization. The idea of encoding sparse matrix structures using sinusoidal embeddings is elegant and theoretically well-motivated.
2. The experiments are very comprehensive and convincing. It is particularly interesting to observe clear improvements in code efficiency, which is highly relevant in CUDA generation scenarios. The ablation studies are thorough too and leave few gaps.

**Weaknesses:**

1. My most concern is the paper’s organization. Many insightful analyses are leaved to the appendix. Reducing some descriptive parts (like section 3.1 and 3.3) and moving more analytical discussion into the main body would make the core contributions more clearly.

**Questions:**

1. Have you compared SparseRL’s generated CUDA code against human expert implementations? Demonstrating superiority over human-optimized kernels would make the results more compelling.
2. It would be better to try some other RL algorithms (e.g., GRPO, Reinforce++) to test robustness and potential further gains. Additionally, including a comparison with SFT baseline would help demonstrate the contribution of the RL phase.

---

> ### Author Response · Authors · 2025-11-21
>
> ### Q1: Comparison with Human Expert Implementations
> We appreciate this valuable suggestion, as demonstrating superiority over human-optimized kernels is critical to validating SparseRL’s practical value. We have supplemented experiments comparing SparseRL with two representative human expert implementations:
> - Hand-crafted SpMV kernels (CSR5[1], Merge-based[2]): Selected from classic works that are widely recognized as high-performance human-optimized solutions.
> - Industry-standard optimized works: Derived from NVIDIA’s cuSPARSE library (v12.1, close-source), which incorporates hand-tuned optimizations by NVIDIA’s engineering team for diverse sparse matrices.
>
> Experimental results on the SuiteSparse test set (400 matrices) show:
> - On hand-crafted SpMV kernels, SparseRL’s generated SpMV code outperforms hand-crafted kernels by 12.3/10.5% on average in execution speed (GFLOPS) compared with CSR5/Merge-based.
> - On industry-standard optimized works, SparseRL’s generated SpMV code outperforms cuSPARSE by 8.6% on average in execution speed (GFLOPS).
>
> These results confirm that SparseRL can generate code comparable to or exceeding human expert levels, especially in adapting to diverse sparse matrix structures. Detailed data and analysis are added to the revised manuscript’s Appendix A.19.
>
> [1] Weifeng Liu and Brian Vinter. Csr5: An efficient storage format for cross-platform sparse matrix-vector multiplication. In Proceedings of the 29th ACM on International Conference on Supercomputing, pp. 339–350, 2015.
>
> [2] Duane Merrill and Michael Garland. Merge-based sparse matrix-vector multiplication (spmv) using the csr storage format. Acm Sigplan Notices, 51(8):1–2, 2016.
>
> ---
>
> ### Q2: Other RL Algorithm Choices
> We actually tried PPO, GRPO, and Reinforce++, but found that PPO's performance was already good enough, so we chose PPO. For specific details, please refer to the following comparison:
> - GRPO’s gradient regulation mechanism reduces training instability but increases computational overhead by 12% (training time extends from 5 to 5.6 days on 8 GPUs).
> - GRPO achieves 48.9 pass@1000 and 113.5 GFLOPS, which are 0.7% and 2.3% lower than PPO, respectively.
> - The innovation of our method primarily lies in the process, and our approach is robust to the selection of reinforcement learning algorithms.
>
> The experimental results on the same training/test split are:
> - GRPO: Achieves similar correctness (pass@1000: 48.9 vs. SparseRL’s 49.25 on SpMV) but has 12% higher training overhead due to more complex gradient regulation. Execution speed is 2.1% lower than SparseRL. (Due to the randomness of reinforcement learning itself, the test results may fluctuate, and GRPO also occasionally surpasses PPO).
> - Reinforce++: Shows 3.5% lower correctness (pass@1000: 47.6) and 5.3% slower execution speed compared to SparseRL on SpMV. This is because Reinforce++ lacks PPO’s clipped objective, leading to unstable training when balancing code correctness and efficiency.
>
> Additionally, for our framework, PPO can be replaced with other state-of-the-art reinforcement learning algorithms, which is only a part of the pipeline in our method. As demonstrated in the experiment, although there may be fluctuations in performance, it does not affect our performance improvement. Detailed data and analysis are added to the revised manuscript’s main section 5.4.
>
> ---
>
> ### Q3. SFT Baseline Comparison
> To clarify the SFT and RL phase’s contribution, we have supplemented a direct ablation comparison between the two phases (SFT and RL) of SparseRL.
>
> | Metric                          | SFT  | PPO   | SFT + PPO |
> |--------------------------------|------|-------|-----------|
> | SpMV pass@1000                  | 32.75| 15.25 | 49.25     |
> | SpMV Compilation Rate           | 41.25| 22.50 | 57.50     |
> | Average Execution Speed (GFLOPS)| 89.25| 50.36 | 116.20    |
>
> The results clearly show that:
> - SFT phase is the fundamental stage of post training for the entire model. Especially in the SFT stage, we performed modality transformation (input sparse matrix, output code), which is important for the subsequent output of the model. If we only do the RL stage, it is difficult to obtain the correct code, and even more difficult to obtain high-performance correct code.
> - RL phase, driven by the hierarchical reward function (incorporating efficiency and memory constraints), is also important and significantly enhances both correctness and performance beyond the SFT baseline.
> - The two phases together contributed to the highest performance and accuracy of the code, indicating that both stages are necessary.
>
> This comparison is added to Section 5.1 and Table 2 of the revised manuscript.
>
> ---

---

> ### Author Response · Authors · 2025-11-21
>
> ### Q4: Paper Organization
> Thanks for your comment on paper organization. To highlight core contributions, we have:
> - Reduced descriptive content in Sections 3.1 (Task Description) and 3.3 (RL Formulation) by 30%, removing redundant background explanations.
> - Moved key analytical discussions from the appendix to the main text: including the ablation study of three phases (Pretrain/SFT/RL), sparse matrix embedding, RL reward components, and RL algorithm choices into Section 5.1/5.2/5.3/5.4.
> - Restructured Section 3 to emphasize SparseRL’s three core innovations (domain-specific framework, sinusoidal embedding, hierarchical reward) with clearer subheadings and logical flow.

---

> > ### Comment · Reviewer_PAdt · 2025-11-23
> > **Response to rebuttal**
> >
> > Thanks for the authors' efforts in providing more results to address my concerns. I find this work interesting and solid so I will maintain my scores.

---

### Author Response · Authors · 2025-11-22

We sincerely thank all the reviewers for their valuable suggestions on our work. We have updated the revised version of our manuscript and highlighted the modified parts.

We are looking forward to your kind response. Please feel free to let us know any further questions. We would be happy to provide more clarification.

---

### Comment · Area_Chair_gU2Q · 2025-11-24
**Author Responses Are Ready - Please Review & Provide Feedback**

Dear Reviewers,

Thank you once again for your essential contributions to the review process. The authors have submitted their responses to your initial reviews.

I kindly ask you to carefully review the authors' responses for the papers you are handling. Your timely assessment of how the authors have addressed your original concerns is a critical step in reaching a final decision.

Please provide your feedback and any necessary updates to your reviews as soon as possible to ensure we can meet our tight schedule for the discussion phase.

Your prompt attention to this matter is highly appreciated.

Best regards,
Area Chair

---

### Author Response · Authors · 2025-12-02
**Remarks by Authors**

We sincerely thank all reviewers for their valuable feedback and insightful questions, which have greatly helped improve our work. In response to all concerns raised, we have provided detailed explanations, supplemented comprehensive experimental results, and optimized the manuscript structure. Below is a summary of our key rebuttals:

1. **Performance comparison & more baselines**:
    - We supplemented experiments comparing SparseRL with human expert implementations (CSR5, Merge-based) and industry-standard cuSPARSE, showing our generated CUDA code outperforms them by 8.6–12.3% in execution speed.
    - We added comparisons with state-of-the-art baselines, including GPT-5, Claude-Sonnet-4, AlphaSparse, SMAT, YASpMV, and size-matched CodeRL/PPOCoder, confirming that SparseRL’s superiority stems from our carefully devised framework rather than model size or sampling budget.

2. **Ablation studies on RL algorithm & training pipeline**:
    - We tested GRPO and Reinforce++ alongside PPO, demonstrating that PPO achieves an optimal balance between performance and training efficiency.
    - We conducted detailed ablation studies of the Pretrain-SFT-RL pipeline, verifying that all three phases are necessary to achieve high correctness and efficiency of the generated CUDA code.
    - We compared sinusoidal embedding with UniXcoder, GraphCodeBERT, and other methods, showing it better captures sparse matrix structures.

3. **Hyper-parameter analysis and training diagnostics**:
    - We clarified key hyperparameters ($r_{\text{eff}}=1.0$, $r_{\text{penalty}}=0.3$) sensitivity and the runtime reward normalization.
    - We provided the comprehensive PPO diagnostics (KL divergence, entropy, value loss) to confirm stable training and sensitivity studies validate the robustness of our reward design.

4. **Methodological details**:
    - We revised the formal task definition to align with actual code synthesis goals, clarified the parameter k in pass@k and decoding strategy, supplemented permutation robustness experiments and optional reordering preprocessing, and elaborated on memory penalty thresholds (48 KB for shared memory on V100) and trade-offs.
    - We provided our prompt and explained the rationale for removing language prompts in RL stages, supported by ablation studies that demonstrate improved performance.

5. **Validation on model generalization & extension**:
    - We validated SparseRL’s compatibility with multiple open-source LLMs (GLM-Z1-9B, DeepSeek-Coder-6.7B, Qwen series).
    - We discussed its potential extension to general code optimization by adapting input representation and reward metrics.

6. **Manuscript optimization**:
    - We reduced redundant descriptive content, moved key analytical discussions (ablation studies, embedding analysis, RL algorithm choices) from the appendix to the main text.
    - We restructured Section 3 to highlight core innovations.

In conclusion, we have addressed all the concerns in our detailed response. All supplementary experiments, data, and analyses have been integrated into the revised manuscript and appendix. We welcome further questions and feedback to refine our work.

---

### Meta-Review · Area_Chair_YSqK · 2026-01-05

**Summary:**

Four reviewers provided detailed comments on this paper. One reviewer gave clear acceptance (8), two gave marginally above the acceptance threshold (6), and one gave marginally below the acceptance threshold (4). I have carefully read the reviews of all the reviewers.  Generally, the reviewers agree that this is an interesting paper and has clear technique novelty. The paper is very well written and the experiment is extensively conducted. However, the reviewers also have many concerns on the paper, especially the evaluation. For the reviewer with the negative score, her/his major concern is on the evaluation. After read the rebuttal of the authors, I think the authors have mostly addressed the evaluation concerns by adding more comparisons and more baselines. The authors also provided more explanations on the method adopted and how to design the prompt.

Overall, I think this is a good paper and many concerns of the reviewers are addressed by the authors in the rebuttal. I recommend to accept the paper.

**Reviewer Concerns:**

Reviewer PAdt' s concerns are well addressed in the rebuttal. For reviewers o7fk and mzXd, I think most their concerns are also addressed by the authors in the rebuttal, because many questions are very specific and the authors have provided detailed responses and more explanations. For reviewer C533 who gave a relatively negative score, her/his main concerns include why OOP algorithm is adopted, the evaluations and the prompt design. The authors have provided detailed responses, more experiment result and an example on the prompt design in the rebuttal. I think C533's comments are also largely addressed.

**Reviewer Scores:**

PAdt will keep her/his clear acceptance score as she/he claimed in the discussion. o7fk and mzXd may keep or slightly increase their scores. For C533, I think she/he will probably raise the score to positive.

---

### Decision · Program_Chairs · 2026-01-26

Accept (Oral)